# Farnesoid X receptor activation by bile acids suppresses lipid peroxidation and ferroptosis

Juliane Tschuck[1], Lea Theilacker [1], Ina Rothenaigner[1], Stefanie A. I. Weiß[1], Banu Akdogan[2], Van Thanh Lam[3], Constanze Müller[4], Roman Graf[1], Stefanie Brandner[1], Christian Pütz[1], Tamara Rieder[5], Philippe Schmitt-Kopplin [4], Michelle Vincendeau[3], Hans Zischka [2,5], Kenji Schorpp[1] & Kamyar Hadian [1] ✉

Ferroptosis is a regulated cell death modality that occurs upon iron-dependent lipid peroxidation. Recent research has identified many regulators that induce or inhibit ferroptosis; yet, many regulatory processes and networks remain to be elucidated. In this study, we performed a chemical genetics screen using small molecules with known mode of action and identified two agonists of the nuclear receptor Farnesoid X Receptor (FXR) that suppress ferroptosis, but not apoptosis or necroptosis. We demonstrate that in liver cells with high FXR levels, knockout or inhibition of FXR sensitized cells to ferroptotic cell death, whereas activation of FXR by bile acids inhibited ferroptosis. Furthermore, FXR inhibited ferroptosis in ex vivo mouse hepatocytes and human hepatocytes differentiated from induced pluripotent stem cells. Activation of FXR significantly reduced lipid peroxidation by upregulating the ferroptosis gatekeepers GPX4, FSP1, PPARα, SCD1, and ACSL3. Together, we report that FXR coordinates the expression of ferroptosis-inhibitory regulators to reduce lipid peroxidation, thereby acting as a guardian of ferroptosis.

Cell death can be initiated in a regulated fashion (regulated cell death–RCD) or happen as uncontrolled necrosis[1,2]. Among the RCD pathways, ferroptosis is a cell death modality, which depends on iron-mediated lipid peroxidation[3]. Within the last decade ferroptosis has become a highly emerging field with implication in diverse disease settings[4–6]. Several ferroptosis-inducing (FIN) agents are able to initiate ferroptosis. Erastin and Imidazole Ketone Erastin (IKE) belong to Class I FINs (inhibitors of system $x_c^-$), while (1 S,3 R)-RSL3 and ML210 belong to Class II FINs (inhibitors of GPX4)[4]. Thus, both of these classes affect the activity of the main ferroptosis gatekeeper axis: system $x_c^-$/glutathione/GPX4. While Class I FINs deplete glutathione, the critical cofactor of GPX4, Class II FINs inactivate GPX4 catalytic activity. In consequence, these molecules prevent the reversion of lethal lipid peroxides (PLOOH) to the non-lethal alcohol forms (PLOH)[7]. In addition to the system $x_c^-$/glutathione/GPX4 axis, the FSP1/ubiquinone axis[8–10], the GCH1/tetrahydrobiopterin/DHFR axis[11,12], and the FSP1/vitamin-K axis[13] are important ferroptosis-inhibitory modules that act independently from GPX4 activity. Moreover, it has been shown that addition and incorporation of monounsaturated fatty acids with the involvement of ACSL3 can counteract ferroptosis[14]. While many ferroptosis-regulatory pathways have been uncovered, there are transcriptional and metabolic processes that remain uncharacterized regarding their involvement in ferroptosis regulation. Notably, there are also chemical radical-trapping antioxidants such as ferrostatins and liproxstatins that inhibit ferroptosis in cells or in vivo[4,5].

[1]Research Unit Signaling and Translation, Helmholtz Zentrum München, Neuherberg, Germany. [2]Institute of Molecular Toxicology and Pharmacology, Helmholtz Zentrum München, Neuherberg, Germany. [3]Institute of Virology, Helmholtz Zentrum München, Neuherberg, Germany. [4]Research Unit Analytical BioGeoChemistry, Helmholtz Zentrum München, Neuherberg, Germany. [5]Institute of Toxicology and Environmental Hygiene, Technical University Munich, School of Medicine, Munich, Germany. ✉e-mail: kamyar.hadian@helmholtz-munich.de

Nuclear receptors (NR) are a family of transcriptional regulators that are activated upon binding of specific agonist ligands such as fatty acids, bile acids, and others. They contain a ligand binding domain and a DNA binding domain and typically act as heterodimers. Acting as a transcription factor, NRs drive expression of distinct target genes[15]. Farnesoid X receptor (FXR) − also known as NR1H4 − belongs to the nuclear receptor family and binds bile acids to get activated[16–18]. FXR forms a heterodimer with Retinoid X receptor (RXR) to initiate gene transcription predominantly in the liver, intestine, and kidney. Importantly, excess FXR activation inhibits bile acid production through upregulation of the transrepressor small heterodimer partner 1 (SHP-1) that represses CYP7A1, a key enzyme for generation of bile acids[19,20]. Besides its role in bile acid homeostasis, FXR is also involved in lipid metabolism and glucose metabolism[19,21]. With these essential functions, alterations in FXR are implicated in many diseases including cholestasis, diabetes, and inflammation[19,21].

In this report, we screened a set of small molecules with known modes of action to search for novel ferroptosis regulators, and identified Turofexorate and Fexaramine (both agonists of FXR) to inhibit lipid peroxidation and ferroptosis. Notably, overexpression or activation of FXR upregulates the ferroptosis-inhibitory proteins FSP1, PPARα, GPX4, SCD1, and ACSL3 to reduce lipid peroxidation induced by FINs.

## Results

### Identification of novel cellular ferroptosis inhibitors
To discover novel cellular regulators of ferroptosis, we screened a library of 3684 small molecules with known targets and modes of action. Ferroptotic cell death was induced by IKE after HT-1080 cells were pre-treated with compounds of the screening library. Cell viability was assessed and every compound that reached a viability higher than the indicated threshold was considered a hit. Among these 16 potential ferroptosis inhibitors were Turofexorate and Fexaramine (agonists of Farnesoid X Receptor), as well as the known ferroptosis inhibitor Ferrostatin-1 (Fig. 1a). To further validate these hits as ferroptosis suppressors, we conducted a 10-point dose-response analysis, which eliminated six compounds from our selection due to lack of reproducible efficacy (Fig. 1b). Additional criteria to consider hits as novel ferroptosis inhibitors were selectivity over other types of cell death. To check ferroptosis-selectivity, we performed apoptosis and necroptosis assays with the verified hit compounds. For apoptosis assay, cell death was induced using Staurosporine; necroptotic cell death was induced via a mixture of TNF-α, the SMAC mimetic LCL-161, and the pan-caspase inhibitor Z-VAD-FMK. Since HT-1080 cells barely undergo necroptosis due to low expression of receptor interacting serine/threonine kinase 3 (RIPK3) (Supplementary Fig. 1; ARCH[4] database)[22], which is essential for necroptosis execution[23], we chose mouse embryonic fibroblasts (MEFs) instead that are both ferroptosis- and necroptosis-sensitive. None of our selected compounds (compound concentrations in Supplementary Data 2) − except ATT − were able to rescue cells undergoing apoptosis or necroptosis (Fig. 1c, d), indicating their selective effect on ferroptotic cell death. We also tested the potency of our selected ferroptosis-inhibitory compounds against FIN56, a class III ferroptosis inducer (FIN). Here, almost all compounds (compound concentrations in Supplementary Data 2) showed a protective effect against FIN56-induced ferroptosis, and the compounds Turofexorate and Fexaramine achieved a significant rescue (Fig. 1e). Finally, we performed flow cytometry using live staining with the BODIPY-C11 lipid peroxidation sensor to evaluate the ability of our hit compounds (compound concentrations in Supplementary Data 2) to protect against lipid peroxidation. Turofexorate and Fexaramine showed inhibition of RSL3- and IKE-induced lipid peroxidation, even more than some of the known ferroptosis suppressors, which we applied as positive controls (Fig. 1f).

Together, using a small molecule compound screen, we identified a number of novel compounds and associated targets that block ferroptosis, including two agonists of Farnesoid X Receptor (FXR). As we pulled two compounds (Turofexorate and Fexaramine) out of the screen modulating the same target (FXR), we pursued with these to decipher the mechanistic details behind their ferroptosis inhibition.

### FXR cooperates with RXR to inhibit ferroptosis
Turofexorate and Fexaramine are known synthetic agonists of FXR, a nuclear receptor that acts as a sensor for bile acids in liver, kidney and small intestine[16–18]. The fibrosarcoma cell line HT-1080, however, expresses only very low levels of FXR (Supplementary Fig. 1; ARCH[4] database)[22]. Interestingly, treatment of HT-1080 cells with Turofexorate or Fexaramine alone led to an increase of *FXR* on mRNA as well as protein levels (Fig. 2a), suggesting that FXR might facilitate the anti-ferroptotic effect of these compounds. In order to verify this hypothesis, we performed a knockdown of *FXR* using an esiRNA. Treatment with IKE at sub-lethal concentration revealed that *FXR*-knockdown significantly sensitized cells towards ferroptosis compared to control cells (Fig. 2b). We also observed that FXR activation by Turofexorate or Fexaramine rescued cells from ferroptotic cell death across a broad concentration range of RSL3 and IKE treatment (Fig. 2c, d). It is known that FXR forms dimers with the Retinoid X Receptor (RXR) in order to bind to response elements and regulate gene expression[15]. To test whether FXR-RXR dimerization is necessary to achieve an anti-ferroptotic effect, we treated HT-1080 cells with the ferroptosis inducers RSL3 or IKE, and added Turofexorate or Fexaramine to rescue cells from ferroptosis. In addition, we treated cells with the RXR antagonist HX 531 (RXRi) (Fig. 2e, f). Importantly, we observed a significant dose-dependent re-sensitization upon addition of the RXR antagonist, indicating that activation of FXR alone is not enough to inhibit ferroptotic cell death, but FXR-RXR need to cooperate to suppress ferroptosis.

To mimic conditions that are closer to physiological conditions, we induced ferroptosis in HT-1080 spheroids and treated them with Turofexorate or Fexaramine. RSL3-induced ferroptosis impaired spheroid formation, whereas co-treatment with the FXR agonists led to intact spheroids identical to DMSO- or Ferrostatin-1-treated controls. High-content-image analysis measuring the ratio of spheroid width to length confirmed significant differences between ferroptotic spheroids and compound-rescued spheroids (Fig. 3a).

Since FXR activation has a more physiological context in liver, we also performed ferroptosis assays in HepG2 cells that display higher levels of FXR (Supplementary Fig. 1, ARCH[4] database)[22]. Hence, treatment of HepG2 cells with Turofexorate or Fexaramine did not further elevate FXR levels as evident from analysis of mRNA and protein levels (Fig. 3b). As previously described, HepG2 cells were less sensitive to ferroptosis induction than HT-1080[11] (Fig. 3c). Still, Turofexorate or Fexaramine dose-dependently inhibited RSL3-mediated ferroptosis in HepG2 cells (Fig. 3c).

To assure target specificity of Turofexorate and Fexaramine, we created a HepG2 FXR KO cell line (Supplementary Fig. 2a). Intriguingly, FXR knockout significantly sensitized HepG2 cells to RSL3-induced ferroptosis when compared to wildtype cells (Fig. 3d), and co-treatment with FXR agonists Turofexorate or Fexaramine could not rescue KO cells from ferroptosis anymore (Fig. 3e). Next, we treated HepG2 cells with the FXR antagonist Guggulsterone (FXRi) together with RSL3 and Turofexorate. FXR inhibition reverted the Turofexorate-ferroptosis-inhibitory-effect, and hence led to more RSL3-induced ferroptosis (Fig. 3f), indicating that the FXR agonist-mediated inhibition of ferroptosis seen in this study is an on-target (FXR) event. Finally, FXR inhibition by Guggulsterone alone sensitized HepG2 cells to ferroptosis induction (Fig. 3g), which fits well with our data that FXR activation (e.g., Turofexorate) inhibits ferroptosis (Fig. 3g).

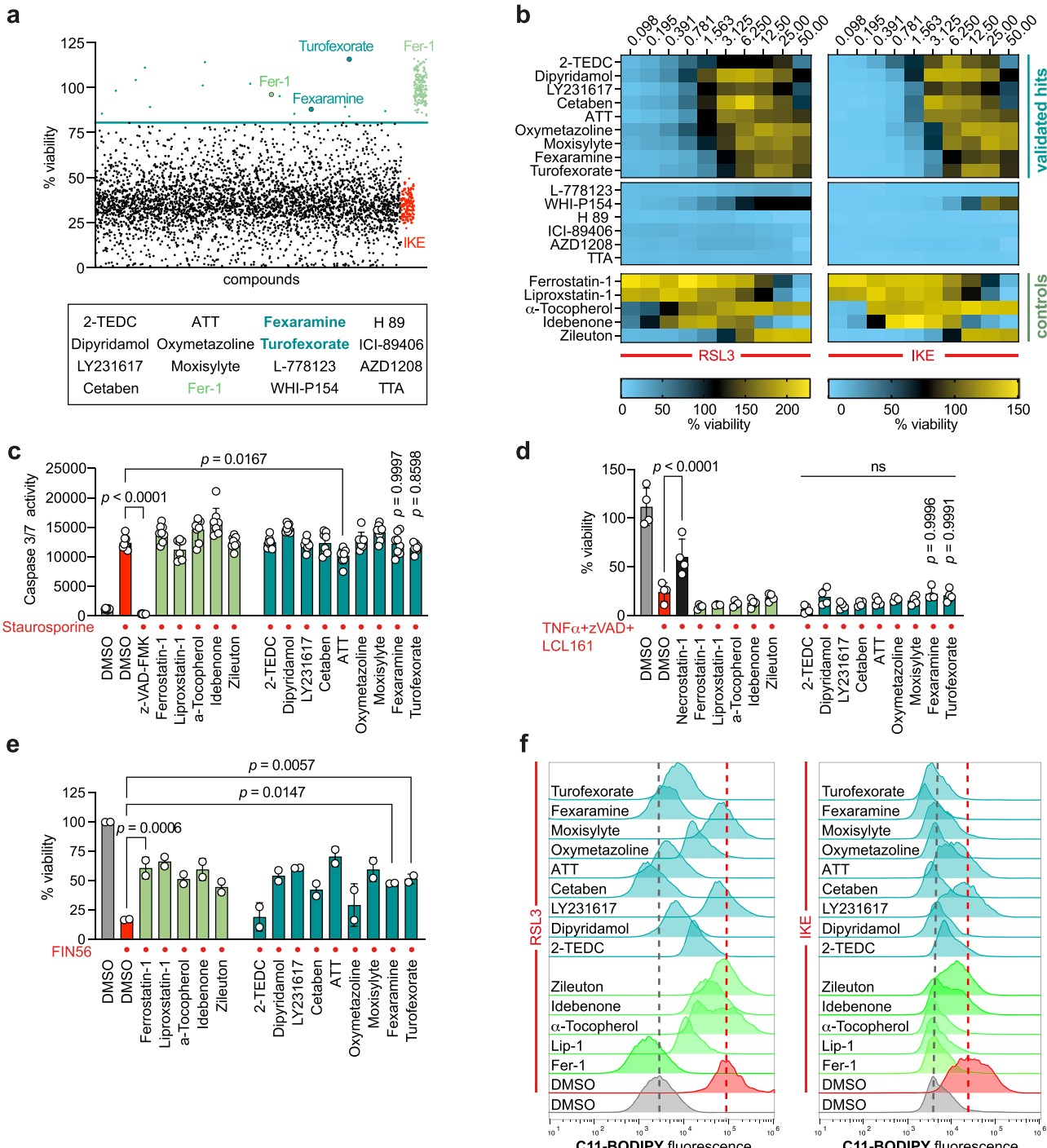

**Fig. 1 | Identification and validation of novel ferroptosis inhibitors. a** Viability screen of 3684 small molecules (see Supplementary Data 1) identified Turofexorate and Fexaramine among 13 other hits as ferroptosis inhibitors after induction with 1.5 μM IKE for 18 h. 2 μM Ferrostatin-1 (Fer-1) was used as positive control. Hit threshold was set as 3x SD from median of compound-treated wells. **b** Dose-response heat map of cell viability with primary hits after ferroptosis induction with 100 nM RSL3 or 1.5 μM IKE for 18 h (*n* = 3 technical replicates). Known ferroptosis inhibitors served as positive controls. **c** Apoptotic cell death was induced with 1 μM Staurosporine in HT-1080 for 18 h, 50 μM Z-VAD-FMK was used as a known apoptosis inhibitor. Compound concentrations can be found in Supplementary Data 2; one-way ANOVA with Dunnett's test. Data are mean ± SD of *n* = 8 technical

replicates. **d** Necroptotic cell death was induced with 20 ng/ml TNFα + 10 μM Z-VAD-FMK + 10 μM LCL161 for 18 h in MEFs, 10 μM Necrostatin-1 was used to inhibit necroptosis. Compound concentrations can be found in Supplementary Data 2; one-way ANOVA with Dunnett's test. Data are mean ± SD of *n* = 4 technical replicates. **e** Turofexorate and Fexaramine inhibit FIN56-induced cell death in HT-1080. Ferroptosis was induced by 200 nM FIN56 for 18 h. Compound concentrations can be found in Supplementary Data 2; one-way ANOVA with Dunnett's test. Data are mean ± SD of *n* = 2 technical replicates. **f** Selected compounds inhibit lipid peroxidation in C11-BODIPY-stained cells. Ferroptosis was induced by 250 nM RSL3 for 2 h, or 2.5 μM IKE for 6 h, compound concentrations can be found in Supplementary Data 2.

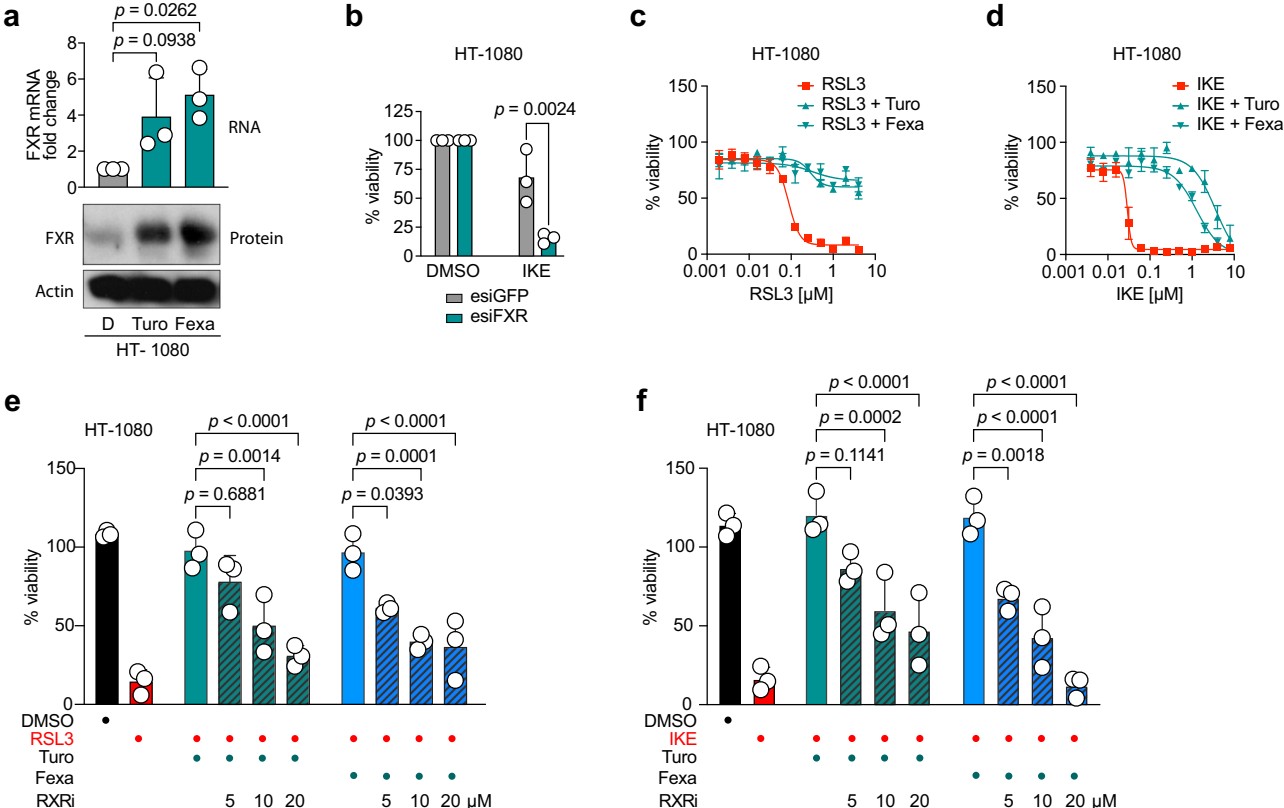

**Fig. 2 | Activation of FXR suppresses ferroptosis in cooperation with RXR.**
**a** Treatment of HT-1080 with 12 μM Turofexorate or Fexaramine for 7 h increases FXR expression. Levels of mRNA were normalized to GAPDH expression. Data are mean ± SD of $n = 3$ biological replicates. Western Blot shown is one representative experiment from $n = 3$. One-way ANOVA with Dunnett's test; D = DMSO
**b** Knockdown of FXR sensitizes cells to ferroptosis. HT-1080 were transfected with 40 nM esiRNA against FXR for 48 h and treated with 1 μM IKE for 18 h. EsiGFP-transfected cells served as a control group. Data are mean ± SD of $n = 3$ biological replicates; one-way ANOVA with Tukey's test. **c**, **d** Turofexorate and Fexaramine are

able to rescue RSL3- and IKE-induced ferroptosis. Cells were treated with 12 μM Turofexorate or Fexaramine and indicated dose range of inducers (RSL3 and IKE) for 18 h. Data are normalized to DMSO and plotted as mean ± SD of $n = 3$ biological replicates. **e**, **f** Dimerization of FXR with RXR is necessary to conduct anti-ferroptotic effect. Ferroptosis was induced with 250 nM RSL3, or 1 μM IKE for 18 h, and 12 μM Turofexorate or Fexaramine was used to inhibit ferroptosis. The RXR transactivation inhibitor HX 531 (RXRi) was added in indicated doses. Data are normalized to untreated control and plotted as mean ± SD of $n = 3$ biological replicates with each 6 technical replicates; one-way ANOVA with Tukey's test.

Altogether, these data demonstrate that FXR activation by Turofexorate or Fexaramine suppresses ferroptosis in 2D and 3D cell culture models. An important requirement for this effect is the cooperative function of FXR with RXR.

### FXR reduces lipid peroxidation
A major hallmark of ferroptotic cell death is peroxidation of polyunsaturated fatty acyl tails (PUFAs). Therefore, we employed a battery of assays to investigate whether FXR activation can reduce lipid peroxidation. First, we immuno-stained 4-Hydroxynonenal (4-HNE), which is a product of lipid peroxidation, and observed via flow cytometry that Turofexorate and Fexaramine significantly reduced 4-HNE levels after ferroptosis induction (Fig. 4a). A further product of lipid peroxidation is Malondialdehyde (MDA), which can be detected using the Thiobarbituric Acid Reactive Substances (TBARS) assay. We performed this assay after induction of ferroptosis in HT-1080 cells and co-treatment with Turofexorate and Fexaramine. The FXR agonists also significantly reduced MDA levels (Fig. 4b). Next, we live-stained HT-1080 and HepG2 cells with the fluorescent lipid peroxidation sensor C11-BODIPY upon ferroptosis induction. In both cell lines, Turofexorate and Fexaramine achieved a significant quenching of BODIPY-related fluorescence, proving that FXR activation inhibits lipid peroxidation (Fig. 4c, d).

Importantly, experiments using C11-BODIPY in a cell-free biochemical setting together with free-radical-producing 2,2'-azobis(2-methyl-propanimidamide) dihydrochloride (AAPH) (Supplementary

Fig. 2b) or the radical 2,2-diphenyl-1-picrylhydrazyl (DPPH) (Supplementary Fig. 2c) revealed that Turofexorate and Fexaramine do not display any antioxidative effect. Also, Turofexorate and Fexaramine did not show any iron chelating capacity (Supplementary Fig. 2d). Hence, we demonstrate that the anti-lipid-peroxidation and anti-ferroptotic effects by Turofexorate and Fexaramine seen in this study are FXR-mediated and not caused by potential chemical properties of these agonists through antioxidation or iron chelating.

Finally, we performed untargeted lipidomics to analyze how FXR activation by Turofexorate impacts lipid composition. Principal component analysis (PCA) showed a clear separation of Turofexorate- and RSL3-treated samples (Fig. 4e). Several PUFA-containing phospholipids (mostly PEs and PCs) were depleted upon RSL3 treatment, and this effect was reverted as a consequence of Turofexorate or Ferrostatin-1 treatment (Fig. 4f), indicating that both inhibitors can counteract lipid peroxidation.

Together, our data reveal that activation of FXR counteracts lipid peroxidation to inhibit ferroptotic cell death.

### FXR upregulates a host of cellular inhibitors of ferroptosis
FXR is a nuclear receptor that can control expression of downstream genes needed for bile acid metabolism[21]. Therefore, we asked whether the anti-ferroptotic effect of FXR activation is mediated through transcriptional control of downstream ferroptosis-relevant genes. To compensate for low *FXR* expression levels in HT-1080 cells and to

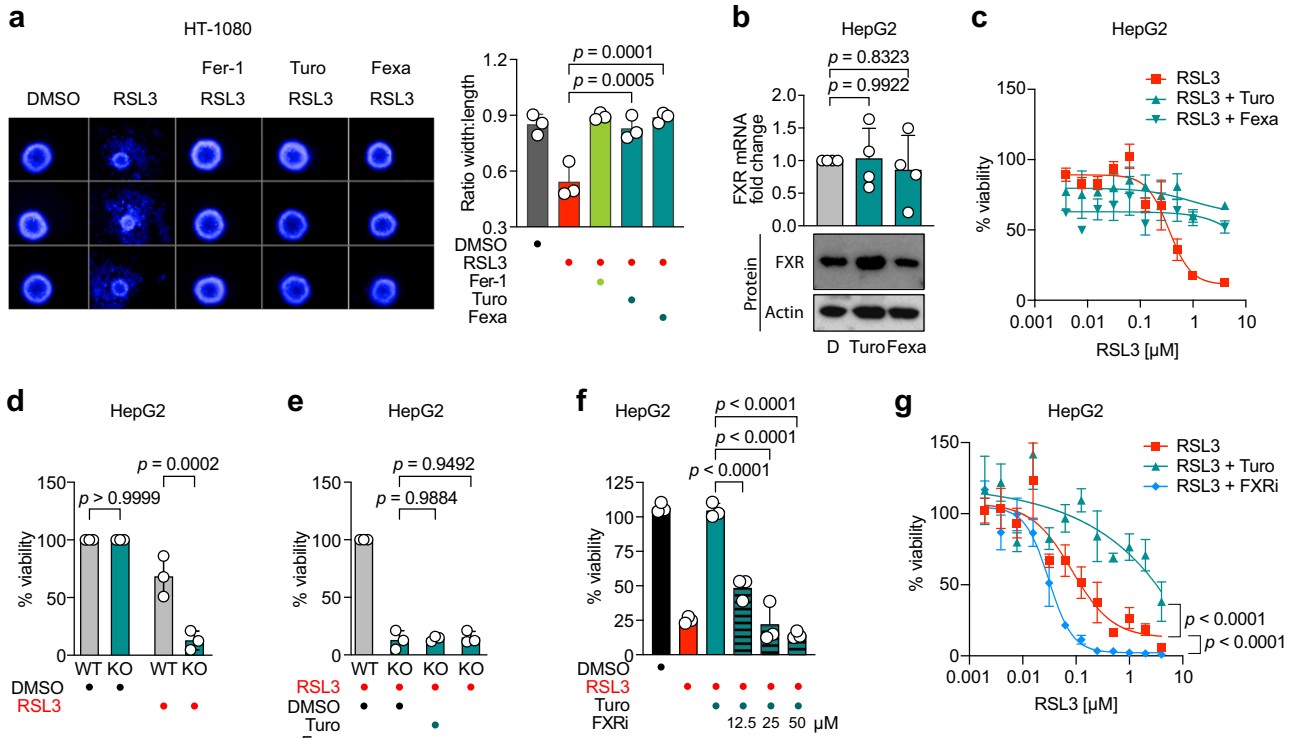

**Fig. 3 | FXR suppresses ferroptosis in 3D spheroid and hepatic cell models. a** HT-1080 3D spheroid models are rescued from ferroptosis by treatment with 12 μM Turofexorate or Fexaramine for 48 h. Ferroptosis was induced by 200 nM RSL3 for 48 h, 2 μM Fer-1 was used as a positive control. Representative spheroid images are shown. Mean data ± SD from $n = 3$ biological replicates with each 8 spheroids were quantified by high-content image analysis; one-way ANOVA with Dunnett's test. **b** HepG2 cells treated with 12 μM Turofexorate or Fexaramine do not show increased levels of FXR. Levels of mRNA were normalized to GAPDH expression. Data plotted are mean ± SD of $n = 4$ biological replicates. Western Blot shown is one representative experiment from $n = 3$. One-way ANOVA with Dunnett's test; D = DMSO. **c** HepG2 cells have lower ferroptosis sensitivity when treated with 250 nM RSL3, but ferroptosis can be rescued by 12 μM Turofexorate or Fexaramine. Ferroptosis was induced for 18 h. Data plotted are mean ± SD of $n = 3$ biological replicates. **d** HepG2 cells with FXR knockout show a higher sensitivity towards

ferroptosis induction. Cells were treated with 125 nM RSL3 for 18 h. Data are normalized to DMSO-treated control and plotted as mean ± SD of $n = 3$ biological replicates; 2-way-ANOVA with Šídák's test. **e** HepG2 cells with FXR KO could not be rescued from ferroptosis by treatment with 12 μM Turofexorate or Fexaramine. Ferroptosis was induced with 125 nM RSL3 for 18 h. Data are normalized to wildtype cells treated with 125 nM RSL3 and plotted as mean ± SD of $n = 3$ biological replicates; one-way ANOVA with Tukey's test. **f** Ferroptosis was induced with 250 nM RSL3 for 18 h, and 12 μM Turofexorate was used to inhibit ferroptosis. The FXR inhibitor Guggulsterone (FXRi) was added in indicated doses. Data are normalized to untreated control and plotted as mean ± SD of $n = 3$ biological replicates; one-way ANOVA with Tukey's test. **g** HepG2 cells can be sensitized towards ferroptosis (250 nM RSL3) by FXR inhibitor Guggulsterone (FXRi). Ferroptosis was induced for 18 h, 12 μM Turofexorate and 50 μM Guggulsterone were used. Data are mean ± SEM of $n = 3$ biological replicates; 2-way-ANOVA with Šídák's test.

further amplify transcriptional effects, we transiently expressed *FXR* yielding robust overexpression (Fig. 5a). Subsequently, quantitative RT-PCR was performed for a number of genes implicated in ferroptosis regulation: *FSP1*, *PPARα*, *GPX4*, *SCD1* and *ACSL3*. These targets were chosen, because the databases Catalogue of Transcriptional Regulatory Interactions (Catrin)[24] and Eukaryotic Promoter Database (EPD)[25,26] predicted that these genes contain transcription factor binding sites for FXR. Besides, these are well-known ferroptosis inhibitory regulators[4,6]. Notably, all five target genes showed a significant increase in mRNA levels under conditions of *FXR* overexpression (Fig. 5b), indicating that these are direct or indirect downstream targets of FXR. We also checked FSP1 and GPX4 on protein level and confirm the upregulation of these genes upon FXR overexpression (Supplementary Fig. 3). Interestingly, when we overexpressed FXR and additionally treated these cells with the FXR inhibitor (FXRi), the inhibition of FXR led to reduction of *FSP1*, *PPARα*, *GPX4*, *SCD1* and *ACSL3* target expression (Fig. 5b), suggesting that the nuclear receptor activity of FXR drives transcription of these ferroptosis gatekeeper genes to reduce lipid peroxidation. To further validate the connection between FXR activity and expression of *FSP1* and *PPARα*, ferroptotic HepG2 cells were rescued with Turofexorate and treated in parallel with inhibitors of FSP1 and PPARα. We demonstrate that small molecule inhibition of FSP1 or PPARα ameliorates the ferroptosis-inhibitory

effect of FXR activation by Turofexorate (Fig. 5c, d), showing that these genes are strongly linked to the FXR-mediated anti-ferroptotic effect. We further explored whether key canonical target genes of FXR (i.e., SHP, NR5A2/LRH-1, FGF19, PLTP, SDC-1)[27] are associated with resistance to ferroptosis inducers by using the Cancer Therapeutics Response Portal (CTRP)[28]. Here, we did not detect any correlation between the canonical FXR target genes and ferroptosis resistance; hence, strengthening the specific function of FXR to upregulate the aforementioned anti-ferroptotic genes.

Next, we investigated whether FXR activation by its endogenous ligands would also rescue cells from ferroptosis. For this, ferroptotic HepG2 cells were treated with chenodeoxycholic acid (CDC), a primary natural bile acid in humans, and obeticholic acid (OC), a semi-synthetic cholic acid derivative. Consistent with our observation using Turofexorate and Fexaramine, the two bile acids were able to significantly inhibit RSL3-induced ferroptosis in HepG2 cells (Fig. 5e). We again performed RT-qPCR in HepG2 cells upon OC and CDC treatment to understand whether endogenous FXR activation had the same effect on the aforementioned FXR target genes. Indeed, treatment with OC and CDC upregulated *FXR* expression and the target genes *GPX4*, *FSP1*, *PPARα*, *ACSL3* and *SCD1* (Fig. 5f). These data show that activation of FXR by bile acids upregulates critical ferroptosis-inhibitory genes to suppress ferroptosis.

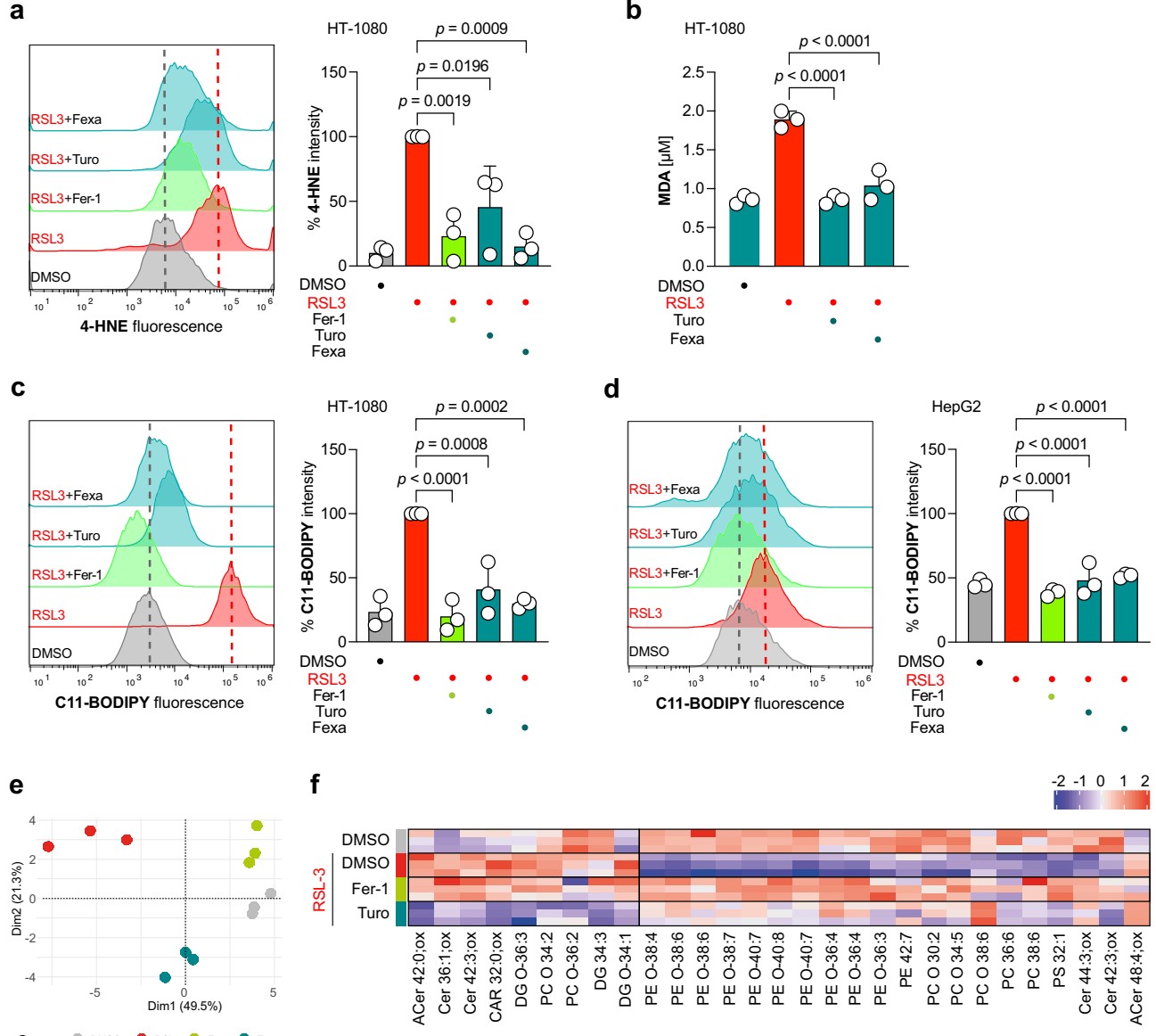

**Fig. 4 | Activation of FXR reduces lipid peroxidation. a** FXR activation by 12 μM Turofexorate or Fexaramine reduces 4-HNE levels. Ferroptosis was induced by 300 nM RSL3 for 2 h. 2 μM Fer-1 was used as positive control. Representative flow cytometry histogram is shown, Data plotted are % intensity of median fluorescence normalized to RSL3 treated cells ± SD of $n = 3$ biological replicates; one-way ANOVA with Tukey's test. **b** FXR activation reduces MDA levels in TBARS assay. Ferroptosis was induced with 250 nM RSL3 for 2.5 h, cells were rescued with 12 μM Turofexorate or Fexaramine. Data are mean ± SD of $n = 3$ biological replicates; one-way ANOVA with Tukey's test. **c**, **d** FXR activation inhibits lipid peroxidation in HT-1080 and HepG2 stained with BODIPY-C11 lipid peroxidation sensor. Cells were treated with 250 nM RSL3 and 12 μM Turofexorate or Fexaramine for 2 h. 2 μM Fer-1 served as a positive control. Representative flow cytometry histograms are shown. Data plotted are % intensity of median fluorescence normalized to RSL3 treated cells ± SD of $n = 3$ biological replicates; one-way ANOVA with Tukey's test. **e** PCA illustrating the phenotypic differences in the cellular pool of HT-1080 cells treated with RSL3, DMSO, RSL3+Turofexorate, and RSL3+Ferrostatin-1 ($n = 3$ technical replicates). **f** HT-1080 cells treated with 200 nM RSL3 for 2 h show modified lipids. PUFA (polyunsaturated fatty acid) containing phospholipids are the predominate drivers. We observed a decrease in relevant highly unsaturated phospholipid species during ferroptosis (RSL3) initiation. Ferrostatin-1 (2 μM) and Turofexorate (12 μM) protected these lipids ($n = 3$ technical replicates). ACer Acylceramide, Cer Ceramide, CAR Carnitine, DG Diacylglycerol, PC Phosphatidylcholine, PE Phosphatidyletha-nolamine, PS Phosphatidylserine.

We next elucidated the effect of FXR activation in ex vivo primary mouse hepatocytes. We induced ferroptosis with RSL3 and co-treated the primary hepatocytes with Turofexorate and Fexaramine, or bile acids (Fig. 6a). Notably, primary mouse hepatocytes were markedly more resistant to ferroptosis induction when compared to HT-1080, which could be attributed to higher FXR levels as one mechanistic possibility. Still, similar to our experiments in the cell lines HT-1080 and HepG2, FXR activation by Turofexorate, Fexaramine, as well as the bile acids OC and CDC led to a significant inhibition of ferroptotic cell death (Fig. 6a). We also evaluated expression levels of *FSP1* and *PPARα*

in primary hepatocytes upon FXR activation and could see an increase in their expression upon Turofexorate and Fexaramine treatment (Fig. 6b).

To further strengthen our hypothesis that FXR is a ferroptosis regulator, we generated another physiologically relevant cell model by differentiating human induced pluripotent stem cells (iPSCs) into functional human hepatocytes (Fig. 6c) expressing essential hepatocyte markers (ALB, ABCC2, CYP1A1, HNF4A, and RXRA) (Supplementary Fig. 4). We then induced ferroptosis with RSL3 in these hepatocytes and performed live cell staining with Propidium iodide.

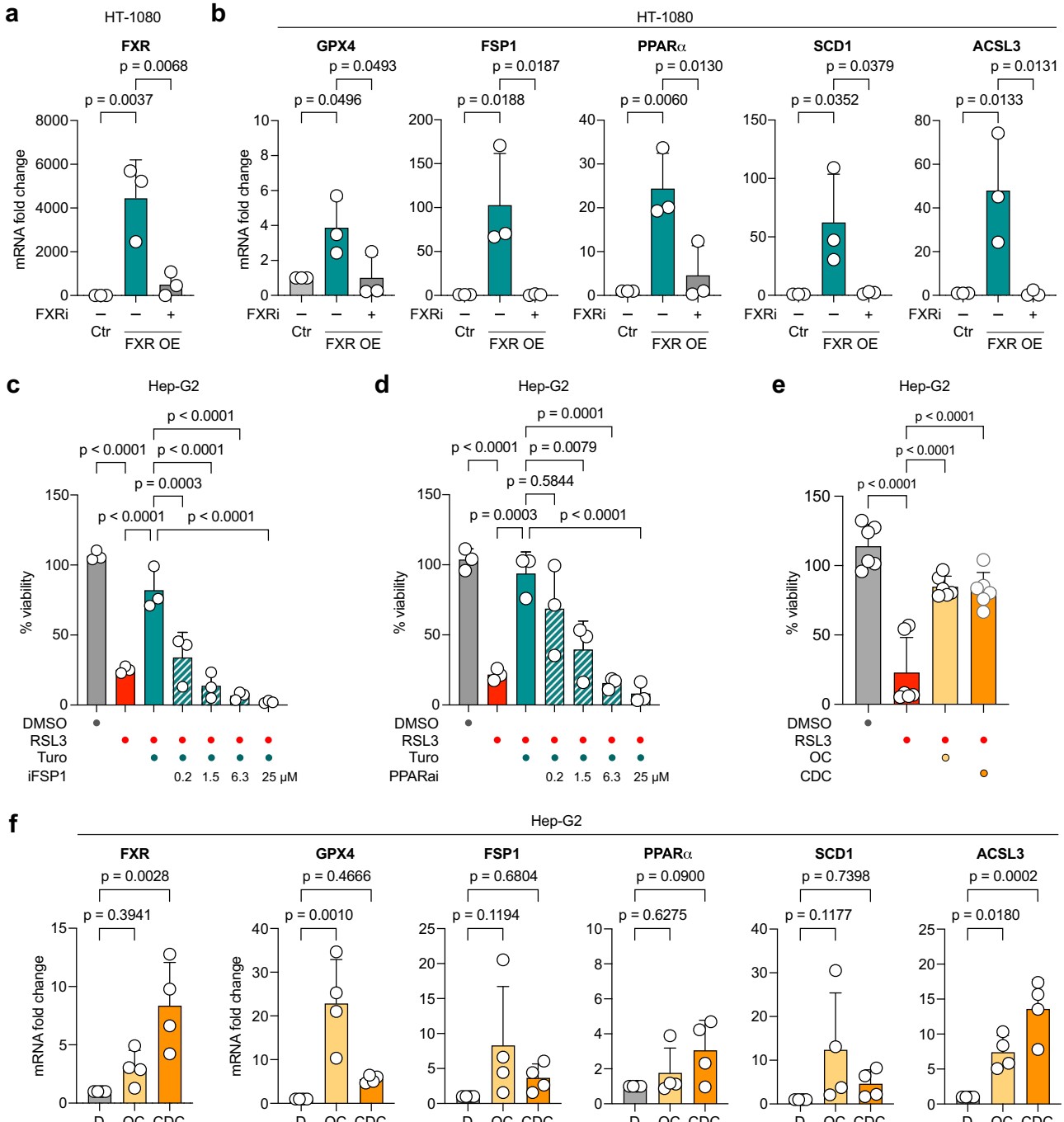

**Fig. 5 | FXR activation induces expression of ferroptosis-inhibitory genes.**
**a** Overexpression of FXR in HT-1080. Levels of mRNA were normalized to GAPDH mRNA. FXR levels are reverted to control levels upon treatment with 50 µM FXR inhibitor Guggulsterone for 24 h. Data are mean ± SD of $n = 3$ biological replicates; one-way ANOVA with Dunnett's test. **b** FXR overexpression leads to elevated mRNA levels of anti-ferroptotic genes GPX4, FSP1, PPARα, ACSL3 and SCD1. Expression levels are normalized to mRNA levels of GAPDH. Upregulated anti-ferroptotic target genes in HT-1080 overexpressing FXR can be reverted to control levels by treatment with 50 µM FXR inhibitor Guggulsterone for 24 h. RP2 was used as a housekeeper gene for normalization. Data are mean ± SD of $n = 3$ biological replicates; one-way ANOVA with Dunnett's test. **c**, **d** Ferroptosis inhibition by FXR activation is reverted when FSP1 or PPARα are inhibited. Ferroptosis was induced by 200 nM RSL3 for 18 h, 12 µM Turofexorate was used to inhibit ferroptosis. FSP1 was inhibited by iFSP1, PPARα was inhibited by GW6471 (PPARαi). Data are normalized to untreated control and plotted as mean ± SD of $n = 3$ biological replicates; one-way ANOVA with Tukey's test. **e** Endogenous FXR activation by bile acids inhibits ferroptosis in HepG2 cells. Cells were treated with 65 nM RSL3, 20 µM Chenodeoxycholic acid (CDC) and 2 µM Obeticholic acid (OC) for 5 h. Data are mean ± SD of $n = 6$ biological replicates; one-way ANOVA with Dunnett's test. **f** Levels of FXR and anti-ferroptotic genes GPX4, FSP1, ACSL3 and SCD1 are elevated after bile acid treatment of HepG2. Cells were treated with 20 µM Chenodeoxycholic acid (CDC) or 2 µM Obeticholic acid (OC) for 8 h. Expression levels are normalized to mRNA levels of GAPDH. Data are mean ± SD of $n = 4$ biological replicates; one-way ANOVA with Dunnett's test; D = DMSO.

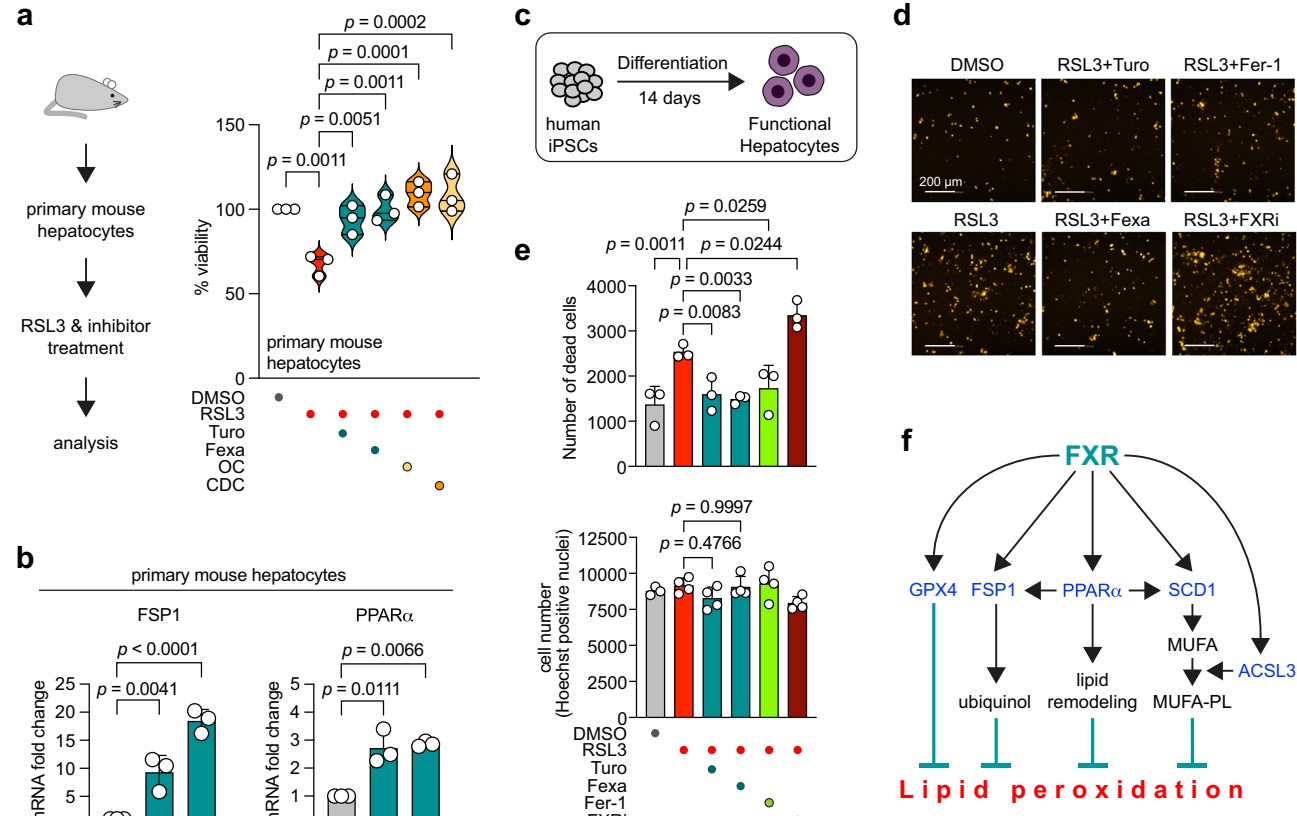

**Fig. 6 | FXR activation inhibits ferroptosis in primary mouse hepatocytes and human iPSC-derived hepatocytes. a** Activation of FXR inhibits RSL3-induced ferroptosis in primary mouse hepatocytes. Isolated cells were treated with 1 µM RSL3 and 12 µM Turofexorate/Fexaramine, 2 µM OC or 20 µM CDC for 18 h; $n = 3$ biological replicates; one-way ANOVA with Dunnett's test. **b** Levels of FSP1 and PPARα are elevated after 12 µM Turofexorate and Fexaramine treatment of primary mouse hepatocytes. Expression levels are normalized to mRNA levels of GAPDH. Data are mean ± SD of $n = 3$ biological replicates; one-way-ANOVA with Dunnett's test. D = DMSO. **c** Induced pluripotent stem cells were differentiated into functional hepatocytes and treated with RSL3 to generate a human primary hepatocyte cell model. **d** Ferroptosis was induced in differentiated hepatocytes (less sensitive) by

treatment with 1 µM RSL3 for 6 h. To rescue cells from ferroptosis, cells were co-treated with 12 µM Turofexorate or Fexaramine or 2 µM Ferrostatin-1. To sensitize cells to RSL3-induced ferroptosis, cells were treated with 50 µM FXR inhibitor (FXRi). Hepatocytes were stained with 5 µg/ml Propidium iodide for live/dead discrimination; scale bar is 200 µm. **e** Quantification of PI-stained hepatocytes. Cotreatment with Turofexorate or Fexaramine rescues cells from ferroptosis, whereas inhibition of FXR sensitizes to cell death. Normalization of cell numbers was done by Hoechst staining. Data are mean ± SD of $n = 3–4$ biological replicates; one-way ANOVA with Dunnett's test. **f** Graphical summary of ferroptosis inhibition by FXR activation.

These data showed a significant reduction of cell death when ferroptotic hepatocytes were co-treated with Turofexorate or Fexaramine, whereas treatment with the FXR inhibitor sensitized cells to RSL3-induced ferroptosis (Fig. 6d, e).

To conclude our findings, we show that FXR activation in cooperation with RXR leads to increased expression of FSP1, PPARα, GPX4, ACSL3 and SCD1, which in turn inhibit lipid peroxidation by different modes of action including antioxidant activity or lipid remodeling (Fig. 6f).

## Discussion

In this study, we conducted a small molecule screen to discover novel ferroptosis-inhibitory pathways and regulators. Among other promising targets, we found that agonistic activation of Farnesoid X Receptor (FXR) by bile acids or the small molecules Turofexorate and Fexaramine selectively rescued cells from ferroptotic cell death by competing lipid peroxidation. This effect was achieved by the molecular function of FXR as a nuclear receptor to transactivate gene expression of various cellular regulators implicated in ferroptosis inhibition. Here, we show that overexpression or activation of FXR upregulates mRNA levels of *FSP1*, *PPARα*, *GPX4*, *ACSL3*, and *SCD1*, and inhibition of FXR reverted the transcriptional upregulation of the target genes. Notably, transcriptional activation of these regulators by FXR is achieved in

cooperation with the Retinoid-X-Receptor (RXR). We also verified increased GPX4 and FSP1 protein levels upon FXR activation. These data are in agreement with a recent study, which indicated that activation of FXR upregulates a set of genes to protect from cisplatin-induced acute liver injury[29]. While activation of FSP1 by FXR overlaps with our study, there are also differences in the gene sets activated by FXR to suppress ferroptotic cell death between the recent[29] and our study, indicating a larger transcriptional rewiring by FXR to block ferroptosis. Together, these two independent studies demonstrate an important role of FXR activation in ferroptosis regulation.

Besides FXR-mediated overexpression of FSP1, which inhibits ferroptosis by generating ubiquinol[8,9], FXR also elevated levels of PPARα — a transcription factor by its own and a master regulator of lipid homeostasis[30]. It has recently been shown that PPARα can upregulate FSP1 transcription[30]. Moreover, another study revealed that PPARα upregulates GPX4 trascription[31], the major gatekeeper of lipid peroxidation and ferroptosis[4,6], which alleviates in vivo iron overload-induced ferroptosis in mouse liver. Therefore, FXR activation seems to have a dual effect on transcriptional activation of ferroptosis regulators: (1) it can activate their transcription directly and (2) it upregulates PPARα that in turn promotes transcription of some of the target genes described above, including the key ferroptosis regulators FSP1 and GPX4. However, we cannot determine the exact amount of

target gene activation by FXR versus PPARα, a detail that is yet to be elucidated in future studies. Importantly, we show that knockout or inhibition of FXR sensitizes HepG2 cells to ferroptotic cell death, and that pharmacological inhibition of FSP1 or PPARα reverts the inhibitory effect of FXR activation; thus, demonstrating the causal relationship between these regulators. We also reveal FXR-associated induction of ACSL3 and SCD1 transcription, both enzymes essential for the generation of monounsaturated fatty acid (MUFA)-containing phospholipids, lipids that acts anti-ferroptotic[14]. Importantly, we did not detect any correlation of canonical FXR target genes to ferroptosis resistance using the Cancer Therapeutics Response Portal (CTRP)[28]. However, it has been shown that NR5A2/LRH-1 (Liver receptor homolog 1) function can have anti-ferroptotic effects in breast cancer[32], and PLTP (Phospholipid transfer protein) may increase anti-oxidative activity in the brain[33]. Thus, while not established as essential anti-ferroptotic genes, some canonical FXR target genes may in part contribute to the anti-ferroptosis activity of FXR activation. Together, we conclude that FXR-mediated upregulation of *FSP1, PPARα, GPX4, ACSL3 and SCD1* is the core mechanism to protect cell membranes from lipid peroxidation during ferroptosis (Fig. 6f).

We detected the ferroptosis-inhibitory effect of FXR in different cell lines, as well as in 3D-spheroid cultures, ex vivo primary mouse hepatocytes and human iPSC-derived hepatocytes. This suggests that activating FXR may be a promising strategy to treat ferroptosis-related degenerative diseases in distinct tissues with high FXR levels. Interestingly, we were also able to revert the ferroptosis-protecting effect of FXR by pharmacologically inhibiting FXR, which opens additional avenues to sensitize certain tumor cells towards ferroptosis induction.

Together, our study shows that the nuclear receptor FXR is a potent transcriptional regulator of ferroptosis by upregulating a number of ferroptosis-inhibitory genes to suppress lipid peroxidation and ferroptosis.

## Methods
### Cell lines and culture conditions
Cell lines: Human fibrosarcoma cell line HT-1080, human hepatocyte carcinoma cell line HepG2 and immortalized mouse embryonic fibroblasts (MEF). MEFs were a gift from Daniel Krappmann, HT-1080 and HepG2 were purchased from ATCC. All cell lines were grown in Dulbecco's Modified Eagle's medium (Thermo Fisher Scientific, 41966-029) containing 10% fetal bovine serum (FBS, Thermo Fisher Scientific), 1% Penicillin-Streptomycin (Thermo Fisher Scientific) and 1% non-essential amino acids (MEM NEAA, Thermo Fisher Scientific). Cells were maintained at 37 °C and 5% $CO_2$.

### Animal experiments
All animal experiments were conducted under the ethical guidelines for the care and use of laboratory animals of the Helmholtz Munich. 41-week-old male C57BL/6 mice (WT and hetero) were maintained on a normal diet from Altromin (*ad libitum*) and sacrificed via cervical dislocation. Primary hepatocytes were isolated by collagenase digestion of livers from mice and all steps were performed according to "Protocol for Primary Mouse Hepatocyte Isolation"[34]. Hepatocytes were seeded into 6-well (300,000 per well) and 384-well microplates (1,000 per well, CulturPlate, PerkinElmer) precoated with collagen (rat tail, Gibco) for 24 h before the isolation.

### Hepatocyte differentiation
Induced pluripotent stem cells (iPSCs) were differentiated into hepatocytes according to the previously published protocol[35]. In brief, cells were seeded into growth factor-reduced Matrigel-coated 6-well plates (2 million cells per well) in STEMdiff definitive endoderm basal medium (StemCell Technologies) containing Supplements A and B and cultivated for one day. From day 2 to 4, medium was exchanged daily and only contained Supplement B. At day 5, cells were detached and re-

seeded into 96-well plates with a density of 12,000 cells per well. Medium was changed to Differentiation medium for hepatic specification: High-glucose DMEM/F12 (Gibco); 10% KOSR (Gibco); 1% Glutamin (Gibco); 1% Non-essential amino acids (Gibco); 1% Pen/Strep (Gibco); 100 ng/ml Human Growth Factor (peprotech); 1% DMSO; 10 µM Rock inhibitor Y27632 (Millipore).

From day 7 to 13, medium was changed to Differentiation medium without Rock inhibitor and refreshed daily. At day 14, medium was changed to Cultivation medium, which contained $10^{-7}$ M Dexamethasone (Sigma Aldrich) instead of Human Growth Factor and DMSO. This medium was also changed daily and used during treatments and further analysis.

To assess the level of iPSC differentiation into hepatocytes, expression levels of markers were analyzed on mRNA level by qRT-PCR: ALB, ABCC2, CYP1A1, HNF4A and RXRA.

### Chemicals
Compounds which were considered a hit after library screening were purchased from the following commercial suppliers: 2-TEDC (Tocris Bioscience), Anethole-trithione (ATT, Selleck Chemicals), Fexaramine (Fexa, MedChemExpress), H 89 HCl (Selleck Chemicals), Dipyridamol (Focus Biomolecules), Oxymetazoline (Santa Cruz Biotechnology), Turofexorate Isopropyl (XL335/Turo, TargetMol), ICI-89406 (Santa Cruz Biotechnology), LY231617 (Tocris Bioscience), Moxisylyte (Santa Cruz Biotechnology), L-778123 (TargetMol), AZD1208 (Tocris Bioscience), Cetaben (Santa Cruz Biotechnology), WHI-P154 (Selleck Chemicals) and Tetradecylthioacetic acid (TTA, Sigma-Aldrich). For hit validation experiments, following known ferroptosis inhibitors were purchased: Ferrostatin-1 (Fer-1, Sigma-Aldrich), Liproxstatin-1 (Lip-1, Sigma-Aldrich), a-Tocopherol (MedChemExpress), Idebenone (Sigma-Aldrich) and Zileuton (Santa Cruz Biotechnology). The following ferroptosis inducers were purchased: (1 S,3 R)-RSL3 (Sigma-Aldrich), Imidazole Ketone Erastin (IKE, Cayman Chemical) and FIN56 (Cayman Chemical).

Compounds and proteins for apoptosis and necroptosis assays: Staurosporine (Stauro, TargetMol), Z-VAD-FMK (zVAD, TargetMol), Tumor Necrosis Factor alpha (TNF-α, biomol), LCL161 (MedChemExpress) and Necrostatin-1 (Nec-1, BioVision). For inhibition experiments, HX 531 (RXR transactivation inhibitor, Biomol), Guggulsterone E&Z (FXR antagonist, Selleck Chemicals), iFSP1 (FSP1 inhibitor, MedChemExpress) and GW6471 (PPARα antagonist, Selleck Chemicals) were purchased. For experiments with endogenous FXR agonists: Chenodeoxycholic Acid (endogenous bile acid, FXR agonist, LKT Labs) and Obeticholic Acid (cholic acid derivative, FXR agonist, BioVision).

### Compound screening
For the screening, HT1080 cells were seeded in 40 µl medium in 384-well microplates (CulturPlate, PerkinElmer) with a cell number of 750 cells per well using a MultiFlo™ Dispenser (BioTek Instruments). 24 h after seeding, cells were treated with compounds (0.5 µl per well–i.e., 5 µM) or DMSO alone as control. Plate and liquid handling were performed using a HTS platform system composed of a Sciclone G3 Liquid Handler (PerkinElmer) with a Mitsubishi robotic arm (Mitsubishi Electric, RV-3S11) and a Cytomat™ Incubator (Thermo Fisher Scientific). The diverse small-molecule library used for this study is composed of 3684 compounds with known mode of action (dissolved in DMSO, 1 mM stock solution). As positive control, cells were treated with the ferroptosis-inhibitor Ferrostatin-1 (2 µM). 4 h later, ferroptosis was induced using 1.5 µM IKE. After 18 h incubation time cell viability was measured by adding 20 µl CellTiter-Glo 2.0 Reagent (Promega). Luminescence signals of the CellTiter-Glo assay were detected on the EnVision 2104 Multilabel plate reader (PerkinElmer). Signals of the Ferrostatin-1-treated wells were set to 100% activity. For hit selection, a threshold of higher than 3 standard deviations from the median of the compound-treated population was set.

## Cell viability assays and dose-response curves

Cells were seeded in a 384-well plate (CulturPlate, Perkin Elmer). Due to their different growth rates, HT-1080 were seeded at 750 cells per well, and HepG2 at 1000 cells per well. After 24 h, treatments with ferroptosis inducers and respective inhibitors were conducted. For dose-response curves, compounds were added in a 10- or 12-point serial dilution. Viability was assessed by adding CellTiter-Glo 2.0 Reagent (Promega) directly into each well and measuring luminescence in an EnVision 2104 Multilabel plate reader (PerkinElmer).

## Apoptosis assay

HT-1080 were seeded into a 384-well plate with a density of 750 cells per well and incubated for 24 h. Apoptosis was induced by Staurosporine before the indicated compounds or zVAD-FMK were added. After 18 h, Caspase 3/7 activity was measured by adding Caspase-Glo 3/7 Assay Reagent (Promega), incubating for 45 min at room temperature and measuring luminescence in an EnVision 2104 Multilabel plate reader (PerkinElmer).

## Necroptosis assay

MEF were seeded into a 384-well plate with a density of 750 cells per well and incubated for 24 h. Necroptosis was induced with a mixture of TNFα, zVAD-FMK and LCL161 before the indicated compounds or Necrostatin-1 were added. After 18 h, cell viability was measured by adding CellTiter-Glo 2.0 Reagent (Promega) directly into each well and measuring luminescence in an EnVision 2104 Multilabel plate reader (PerkinElmer).

## Spheroid formation and imaging

HT-1080 cells were seeded into a 96-well Round Bottom Ultra Low Attachment Microplate (Corning costar 7007) with a density of 2000 cells per well. After 48 h of growth, spheroids were treated with RSL3 and Ferrostatin-1 or Turofexorate/Fexaramine and incubated for additional 48 h. Staining was performed by adding Hoechst 33342 (Sigma) in a 1:10,000 dilution directly into the wells and incubating for 1 h. Images were taken with an Operetta high-content system and analyzed with Columbus software (PerkinElmer). For analysis, spheroids were detected as "Image region" and morphology properties were calculated (e.g., roundness).

## Staining of iPSCs-derived hepatocytes

On day 14 of differentiation, cells were treated with indicated compounds for 6 h, medium was removed and the cells were washed twice with PBS +/+ (containing $Mg^{2+}$ and $Ca^{2+}$, Sigma Life Sciences). By adding 4% PFA with 0,1% Triton-X100, cells were fixed and permeabilized. After three washing steps with PBS + 0,1% Triton-X100, cells were incubated for 1 h at room temperature for blocking with 5% normal goat serum (Thermo Fisher Scientific). Cells were again washed three times with PBS + 0,1% Triton-X100 and incubated with Hoechst 33342 (Sigma, 1:10,000) for 20 min at room temperature. Imaging was performed using an Operetta high-content system and images were analyzed with the Columbus software (PerkinElmer).

For live/dead discrimination, cells were live-stained with 5 µg/ml Propidium iodide (Invitrogen) after 6 h of compound treatment and imaged with an Operetta high-content system. Image analysis was done with the Columbus software (PerkinElmer).

## FXR knockdown

HT-1080 were seeded into 12-well plates with a density of 100,000 cells per well. After 24 h, medium was exchanged with 900 µl of fresh growth medium before cells were transfected with 40 nM of esiRNA against human *FXR* (MISSION esiRNA, Sigma, EHU133161). EsiRNA was diluted in Opti-MEM I Reduced Serum Medium (Thermo Fisher), and 3 µl Lipofectamine RNAiMAX Reagent (Invitrogen) was used as a transfection reagent. As a negative control, esiRNA against *EGFP* (MISSION esiRNA, Sigma, EHUEGFP) was transfected. After transfection, cells were incubated for 10 min at room temperature, before they were incubated again at 37 °C. Medium was changed after 6 h, cells were rinsed with PBS before fresh medium was added. For ferroptosis assays, cells were harvested 48 h after transfection using 0.05% Trypsin-EDTA (Thermo Fisher) and seeded into a 384-well plate with a density of 750 cells per well. After 6 h, knock-down and control cells were treated with IKE for 18 h and viability was measured by adding CellTiter-Glo 2.0 Reagent (Promega) into each well and measuring luminescence in an EnVision 2104 Multilabel plate reader (PerkinElmer). Successful knock down was examined by RT-qPCR.

## Generation of FXR-KO cell lines using CRISPR-Cas9

For generation of a stable HepG2-FXR-KO cell line, a 6-well plate with 250,000 cells per well was seeded. After 24 h of incubation, cells were transfected with TrueCut™ Cas9 protein (Invitrogen), three different TrueGuide™ Synthetic guideRNAs (Invitrogen) against FXR, Lipofectamine™ Cas9 Plus Reagent and Lipofectamine™ CRISPRMAX™ Reagent according to the protocol published by Invitrogen. A non-targeting guideRNA sequence (TrueGuide™ sgRNA Negative Control, invitrogen) was transfected as well. Cells were incubated for 2 days after transfection without medium change. After incubation, all transfected cells were pooled, expanded for 1 week and then seeded into 96-well plates for generation of single cell clones. Clones were tested for FXR expression via qRT-PCR.

## FXR overexpression

HT-1080 were seeded in 6-well plates with a density of 200,000 cells per well. After 24 h of growth, cells were transfected with a vector expressing GFP-tagged human FXR (NM_001206979, OriGene) by using X-tremeGENE HP DNA Transfection Reagent (Sigma-Aldrich). For each well, 2 µg of plasmid were diluted in 200 µl Opti-MEM I Reduced Serum Medium (Thermo Fisher Scientific), before 6 µl of the Transfection Reagent was added. The mixture was incubated for 15 min at room temperature and then added dropwise into every well. As a negative control, cells were only treated with Transfection Reagent without plasmid. 48 h after transfection, cells were examined for GFP expression in an EVOS FL fluorescence microscope (Thermo Fisher Scientific) and harvested with 0.05% Trypsin-EDTA for further experiments.

## RT-qPCR

Total RNA was isolated using the Monarch Total RNA Miniprep Kit (New England BioLabs) according to the manufacturer's instructions. Complementary DNA was synthesized by using the Maxima H Minus First Strand cDNA Synthesis Kit (Thermo Fisher Scientific): after eliminating genomic DNA, oligo $(dT)_{18}$ primer and random hexamer primer were used to generate first-strand cDNA. The incubation time at 50 °C was prolonged to 30 min if template quantities were greater than 1 µg. For RT-qPCR the PowerUp SYBR Green Master Mix (Thermo Fisher Scientific) was used in a LightCycler480 (Roche). Gene expression levels were normalized to *GAPDH* expression and quantified with the delta delta Cp method. List of primers used in RT-qPCR is in Supplementary Data 3.

## Western Blot

HT-1080 or HepG2 were seeded into a 6-well plate with a density of 200,000 cells per well. After 24 h, cells were treated with Turofexorate or Fexaramine for 8 h. Treatment with DMSO was used as a control. Cells were harvested using 2X Roti-Load (Roth) and a cell scraper, before they were lysed with an ultrasonic processor (Hielscher UP200S). Protein denaturation was achieved by incubation of the samples at 95 °C for 5 min. After brief centrifugation, lysate was run on a NuPAGE™ 4–12% Bis-Tris gel (Invitrogen) in 1X MOPS SDS Running Buffer (Invitrogen) and proteins were blotted onto a PVDF membrane

using a semi-dry transfer system. Membrane was blocked with 5% milk powder in TBS-Tween for 30 min at room temperature and incubated in primary antibody (rabbit anti-NR1H4/FXR antibody, ab228949, Abcam; or mouse anti-β-Actin (C4), sc-47778, Santa Cruz Biotechnology; rabbit anti-FSP1 (AMID) antibody, PA5-103183, Thermo Fisher Scientific; or rabbit anti-GPX4 antibody, ab125066, Abcam) overnight at 4 °C. Primary antibodies were diluted in 2.5% milk powder in TBS-Tween (anti-NR1H4/FXR 1:500; anti-β-Actin (C4) 1:500; anti-FSP1 1:1000 and anti-GPX4 1:1000). Afterwards, the membrane was washed 3 × 5 min in TBS-Tween, incubated in secondary antibody (HRP-conjugated AffiniPure donkey anti-mouse IgG, 715-035-150 or HRP-conjugated AffiniPure donkey anti-rabbit IgG, 711-035-152, Jackson ImmunoResearch); diluted 1:7,500 in milk-TBS-Tween) for 1 h at room temperature and washed 3 x 5 min in TBS-Tween. Chemiluminescence was detected using Western Lightning ECL Pro (PerkinElmer). Bands were quantified using ImageJ and normalized to β-Actin signal. Western Blot full scan images are shown in Supplementary Fig. 5.

### Flow cytometry

HT-1080 were seeded either in 6-well plates (200,000 cells/well) or 12-well plates (100,000 cells/well) and incubated for 24 h. Cells were treated with RSL3 for 2 h or IKE for 6 h together with the indicated compounds before BODIPY 581/591 C11 (Thermo Fisher) was added in a final concentration of 2 μM per well and incubated for 30 min at 37 °C. Cells were harvested with 0.05% Trypsin-EDTA (Thermo Fisher) before 2 wash steps were conducted with PBS by spinning down the cells at 500 x g, 4 °C for 5 min. For flow cytometry measurement (Attune acoustic flow cytometer, Applied Biosystems) the cells were resuspended in 300 μl PBS and 10,000 events per condition were analyzed in the BL-1 channel. For immunostaining with anti-4-Hydroxynonenal antibody (4-HNE, ab46545, Abcam) cells were treated with RSL3 and Ferrostatin-1 or Turofexorate/Fexaramine for 2 h before they were harvested with trypsin, centrifuged 5 min at 500 x g, resuspended in 10% normal goat serum (Thermo Fisher Scientific) and incubated for 30 min on ice. After that, cells were stained with anti-4-HNE antibody (1:50 dilution in 1% BSA in PBS) for 1 h on ice. Three washing steps with PBS and centrifugation at 500 x g for 5 min were conducted before cells were resuspended in anti-rabbit Alexa 488 antibody (1:200 dilution in 1% BSA in PBS, A32731, Thermo Fisher Scientific) and incubated for 30 min on ice. After two washing steps with PBS, cells were resuspended in 300 μl PBS and fluorescence of 10,000 events per condition was measured using the BL-1 channel of an Attune acoustic flow cytometer (Applied Biosystems). Histograms and intensities were analyzed with FlowJo v10.8.1 Software (BD Life Sciences). The gating strategy for all flow cytometry analyses is shown in Supplementary Fig. 6.

### TBARS assay

HT-1080 cells were seeded into 150 mm dishes with a density of 2 million cells per dish. After 48 h of growth, cells were treated with RSL3 and Ferrostatin-1 or Turofexorate/Fexaramine for 2.5 h before they were harvested using 0.05% Trypsin-EDTA (Thermo Fisher Scientific). Cells were counted and cell number was adjusted to the sample with the lowest cell count. TBARS Assay was performed according to the user manual of the TBARS (TCA Method) Assay Kit (Cayman Chemical, 700870). Malondialdehyde (MDA) levels were obtained by detecting fluorescence at 530 nm/550 nm in an EnVision 2104 Multilabel plate reader (PerkinElmer).

### Cell-free BODIPY assay

Turofexorate, Fexaramine, and Ferrostatin-1 were diluted in 150 μl PBS to obtain a final concentration of 25 μM. The same amount of DMSO was diluted in 150 μl PBS to serve as a control. In other tubes, BODIPY 581/591 C11 (Thermo Fisher Scientific) was diluted to 1.875 μM in 150 μl

PBS and 2,2′-Azobis(2-methylpropionamidine) dihydrochloride (AAPH, Sigma) was diluted to 7.5 mM in 150 μl PBS. One sample was generated as DMSO only without AAPH to serve as a non-oxidized control. All three solutions were mixed and incubated in the dark for 30 min at room temperature. 100 μl of each condition were transferred into a black 96-well plate (Greiner Bio-One) and fluorescence was measured at 495 nm/520 nm in an EnVision 2104 Multilabel plate reader (PerkinElmer).

### DPPH assay

To determine the antioxidant capacity of Turofexorate and Fexaramine in a cell-free assay, compounds were diluted to a concentration of 10 mM in DMSO. 5 μl of each compound was added into 1 ml of 0.05 mM 2,2-diphenyl-1-picrylhydrazyl (DPPH, Sigma-Aldrich) in methanol before they were rotated for 10 min at room temperature. Samples were transferred into a clear bottom 96-well plate in quadruplicates and absorbance was determined using an EnVision 2104 Multilabel plate reader (PerkinElmer). As a gold standard antioxidant, Ferrostatin-1 was measured as a positive control.

### Ferrous Iron Chelator assay

The Ferrous Iron Chelating (FIC) Assay Kit (amsbio) was performed according to manufacturer's instructions. 12 μM Turofexorate or 12 μM Fexaramine were added to a working solution of $FeSO_4$, and 100 μM of EDTA were added as a positive control. A sample with DMSO in $FeSO_4$ was used as a negative control. To start the reaction, a working solution of Ferrozine was added to samples and control. After 10 min of incubation at room temperature, absorbance at 562 nm was measured in an EnVision 2104 Multilabel plate reader (PerkinElmer). Measured values were normalized to EDTA and presented as percentage of chelated ferrous iron.

### Lipidomics

HT-1080 were seeded into a 100 mm dish with a density of 2 million cells per dish. After 24 h, cells were treated with RSL3, and Ferrostatin-1 or Turofexorate for 2 h, before they were harvested using Accutase solution (Sigma-Aldrich). Cells were centrifuged at 500 x g for 5 min and washed with PBS twice. Before processing, cell pellets were frozen in liquid nitrogen and stored at −80 °C. Procedures for lipid extraction and global lipidomics profiling using UPLC-MS have been described in ref. 36. Briefly, sample extracts were analyzed using reverse phase chromatography (RP) columns in both positive and negative electrospray modes (maXis, Bruker Daltonics, coupled to an UHPLC Acquity, Waters Corporation). 10 μl were injected. RP was performed on a CORTECS UPLC C18 column (150 mm × 2.1 mm ID 1.6 μm, Waters Corporation) using 10 mM ammonium formate and 0.1% formic acid in 60% acetonitrile/water mixture (A) and 10 mM ammonium formate and 0.1% formic acid in 90% isopropanol/acetonitrile mixture (B) as mobile phase. The gradient was set to 32% B for 1.5 min, followed by an increasing proportion of B to 97% at minute 21 and a plateau for the remaining 4 min with a flow rate of 0.25 ml/min. Column temperature was stable at 40 °C. Raw data were extracted, peak identified and quality control processed using Genedata software (Genedata Expressionist 13.5, Genedata). M/z features were annotated using MassTRIX web server and LIPIDMAPS with a maximal mass error of 0.005 Da. Missing values were imputed by randomly generated minimum values and the data was TIC normalized. 2363 mass features were obtained. We used ferroptosis-related lipid modifications by RSL3-induced ferroptotic cells as a pattern to characterize the most relevant lipids in this cell type and concentrated on annotated lipids in the retention time window of minute 10−20 (597 annotated lipid species). Statistical analysis was performed in R studio (R version 4.1.1). Unit variance scaling and mean centering was applied before statistical testing. A Kruskal-Wallis test for non-parametric variables was performed to identify lipids that show significant change between DMSO,

Ferrostatin-1, RSL3, and Turofexorate. Benjamini Hochberg correction was applied to account for multiple testing.

## Statistical analysis

Statistical analyses were carried out using the GraphPad Prism Software version 9.5.1. All statistical details are included into the figures and legends.

## Reporting summary

Further information on research design is available in the Nature Portfolio Reporting Summary linked to this article.

## Data availability

All data are available in the article and its Supplementary Information as well as in the online source file. Western Blot full scan images are shown in Supplementary Fig. 5. Compound library used in Fig. 1a is in Supplementary Data 1. Data in Supplementary Fig. 1 is available online from ARCH[4] database. Source data are provided with this paper.

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

## Acknowledgements

We thank Helmholtz Munich for institutional funding.

## Author contributions

K.H. conceptualized the study; J.T., L.T., I.R., S.W., B.A., C.M., R.G., S.B., C.P., T.R., V.T.L. performed experiments; J.T., L.T., I.R., S.W., B.A., C.M., R.G., K.S., K.H. analyzed data; K.H., K.S., H.Z., P.SK., M.V. supervised the research; K.H., J.T., K.S. wrote the initial paper draft; all authors read, commented, and approved the manuscript.

## Funding

## Competing interests

J.T. and K.H. are inventors on a patent application involving ferroptosis. The remaining authors declare no competing interests.
