## [Peer Review File · Nature Communications]

Farnesoid X Receptor activation by bile acids suppresses lipid peroxidation and ferroptosisREVIEWER COMMENTS

Reviewer #1 (Remarks to the Author):

This manuscript identified that farnesoid X Receptor activation suppressed ferroptosis through upregulating the ferroptosis-inhibitory regulators FSP1, PPAR α , GPX4, SCD1, and ACSL3 to reduce peroxidized lipids. Modulating FXR activity may be beneficial to overcome ferroptosis-mediated degenerative diseases. This is an interesting but preliminary study.

1. Is it quite certain that the antagonists of FXR not chelate iron or scavenge radicals themselves? Many small molecules can block ferroptosis through off-target mechanisms.
2. FXR activation regulated ferroptosis-inhibitory regulators mRNA expression upregulation, their protein levels increased? Western blots should be done.
3. How does FXR activation regulate ferroptosis-inhibitory regulators?
4. The authors should confirm their conclusion using a specific disease model.

Reviewer #2 (Remarks to the Author):

Comments to author:

Reviewer #1: In this manuscript, Tschuck et al., report the results of the pharmacological high throughput screen aiming to identify novel ferroptosis inhibitors. The study reports a protective effect of Turofexorate and Fexaramide, two Farnesoid X Receptor (FXR) agonists. The authors propose that the mechanism is mediated by the an FXR-dependent upregulation of several antiferroptotic proteins including GPX4, FSP1, SCD1 and PPAR α .

The study is exciting and the effect of FXR as a central regulator of ferroptosis is important and clinically relevant. The observation that the agonist triggers a strong response in FSP1 expression level in primary hepatocytes is particularly exciting.

I only have a few remarks the authors might wish to address to increase the soundness of their work-

- 1) I would strongly encourage the authors to check if FSP1 is also upregulated at the protein level in their model. If available, testing the other regulators would be also advisable.

2) To clearly demonstrate the on-target effect of Turo/Fexa the generation of FXR knockout cell line would be important to unequivocally link the agonist to its target in the context of ferroptosis. Its not unlikely that the response could be mediated independent of FXR, which is fine - and could stimulate subsequent studies to full characterize this.

3) Similarly, it would be advisable to exclude unwanted antioxidant activity of Turo/Fexa (a simple DPPH assay should suffice).

4) Are canonical FXR-inducible genes associated with ferroptosis resistance in other cell lines? Some exploration of databases such as the CTRP or Depmap could offer additional insights -

Reviewer #3 (Remarks to the Author):

These elegant series of studies indicate that FXR/RXR activation results in decrease in lipid peroxidation and inhibition of ferroptosis.

This reviewer has some questions:

1) Please present Figure 4F in a clearer format and explain the lipids that are regulated and their significance.

2) Do these lipid changes also occur in vivo, in models of liver injury associated with ferroptosis?

3) Is there any evidence that inhibition of ferroptosis with ferrostatin-1 or liproxstatin-1 or FXR agonists prevent lipid peroxidation and ferroptosis in vivo?

4) Which of the 5 genes stated in Figure 5B are most specific for ferroptosis?

5) In Figure 5F it would be desirable to compare effects of CDC and OC with FXR Overexpression as in Figure 5B.

Rebuttal Letter

Reviewer #1 (Remarks to the Author):

This manuscript identified that farnesoid X Receptor activation suppressed ferroptosis through upregulating the ferroptosis-inhibitory regulators FSP1, PPAR α , GPX4, SCD1, and ACSL3 to reduce peroxidized lipids. Modulating FXR activity may be beneficial to overcome ferroptosis-mediated degenerative diseases. This is an interesting but preliminary study.

We hope that the manuscript has matured with the new data added during the revision process.

1. Is it quite certain that the antagonists of FXR not chelate iron or scavenge radicals themselves? Many small molecules can block ferroptosis through off-target mechanisms.

To exclude any radical scavenging or iron chelating activity of Turofexorate and Fexaramine, we performed different cell-free assays: A cell-free BODIPY-C11 assay and a DPPH assay showed that Turofexorate and Fexaramine do not act as antioxidants, and an iron chelator assay with EDTA as a positive control revealed no iron chelating capacity for both FXR agonists (**NEW Supplementary Fig. 2 B-D**). With these data, we are reassured that we are not seeing off-target effects.

We also provide new data based on the suggestions of reviewer 2, which show that HepG2 FXR KO cells cannot be rescued from ferroptosis by Turofexorate and Fexaramine, proving on-target activity (**NEW Fig. 3E**).

In addition, we performed FXR inhibitory experiments in HT-1080 cells upon FXR overexpression and observe a reduction in target-gene expression, demonstrating that FSP1, PPAR α , GPX4, SCD1, and ACSL3 upregulation need the activity of FXR as a transcription factor (**NEW Supplementary Fig. 4**).

2. FXR activation regulated ferroptosis-inhibitory regulators mRNA expression upregulation, their protein levels increased? Western blots should be done.

We treated HepG2 cells with the FXR agonists Turofexorate and Fexaramine and observed similar results on protein levels compared to the mRNA levels for selected target genes: upon treatment, the key anti-ferroptotic regulators FSP1 and GPX4 were upregulated on protein level (**NEW Supplementary Fig. 3B**). The same effect could be seen in FXR-overexpressing HT-1080 cells, where increasing protein levels of FXR led to elevated protein levels of GPX4 and FSP1 (**NEW Supplementary Fig. 3A**). Western Blot bands were quantified using ImageJ by normalizing to β -Actin.

3. How does FXR activation regulate ferroptosis-inhibitory regulators?

The Farnesoid X Receptor belongs to the superfamily of nuclear receptors, which act as cellular transcription factors. Upon ligand binding (endogenous bile acids or synthetic small molecules in the case of FXR) in the cytoplasm, the receptor dimerizes with itself or other nuclear receptors (homodimers or heterodimers with *e.g.*, RXR), relocates into the nucleus and binds to DNA response elements in promoter regions of target genes to activate transcription. FXR is primarily known to regulate the expression of genes necessary for bile acid metabolism, but ChIP-Seq-based studies and databases showed FXR binding sites in promoter regions of the investigated

ferroptosis-inhibitory genes including FSP1, PPAR α , GPX4, SCD1, and ACSL3 ((Chong *et al.*, 2010; Xing *et al.*, 2022), FXR-Binding-search-Tool TU Graz).

In our study, we show that FXR dimerizes with RXR to display the anti-ferroptotic effect (**Fig. 2E, F**). We observed the upregulation of FSP1, PPAR α , GPX4, SCD1, and ACSL3 upon FXR overexpression/activation (**Fig. 5B, F; NEW Supplementary Fig. 3A, B**), and conclude that FXR not only regulates genes within bile acid metabolism, but also genes that are essential ferroptosis gatekeepers (FSP1, PPAR α , GPX4, SCD1, and ACSL3), which in turn limit lipid peroxidation. This information is provided in the manuscript.

4. The authors should confirm their conclusion using a specific disease model.

We agree with the reviewer that it is interesting to follow up our conclusions in relevant animal models. However, our study **aimed at underpinning the mechanism** by which FXR activation by agonists counteracts ferroptosis. We were able to provide strong and convincing evidence that activation of FXR orchestrates the transcriptional upregulation of ferroptosis gatekeepers to prevent lipid peroxidation. It is this mechanistic key that is the center of our work.

To deepen this focus on mechanistic insights and their relevance at the cellular level, we provided data on the suppression of ferroptosis in *ex vivo* primary mouse hepatocytes by FXR agonists. Importantly, and to further support our conclusions, we have **now added** another physiologically relevant *in vitro* model, human induced pluripotent stem cells (iPSCs) that were differentiated into functional human hepatocytes as verified by a set of hepatocyte markers (**NEW Supplementary Fig. 5**). Ferroptosis was induced in these hepatocytes by RSL3. Ferroptotic cell death was determined by staining the hepatocytes with propidium iodide (PI) and rescue of cell death by Fer-1. In exact agreement to our previous results, co-treatment with the FXR agonists Turofexorate and Fexaramine decreased ferroptosis, whereas treatment with the FXR inhibitor sensitized cells towards RSL3-induced ferroptosis (**NEW Fig. 6D, E**). Thus, we provide relevant information that ferroptosis inhibition via FXR activation can not only be achieved in hepatocellular carcinoma (HepG2) and HT1080m cells, but also in 3D spheroid models, in *ex vivo* murine hepatocytes, as well as in human iPSC-derived hepatocytes.

Of note, we are pleased to see that *in vivo* relevance of FXR to inhibit ferroptosis has recently been reported (Kim *et al.*, 2022; Cao *et al.*, 2023), clearly supporting our findings and independently confirming that FXR is a master regulator of ferroptosis. Also, one of the FXR target genes, namely PPAR α , has been shown to reduce *in vivo* induced ferroptosis.

Taking our new data and the recent reports together, we believe that the *in vivo* relevance is at hand, and that we would like to refrain from their repetition, as we feel committed to the 3R principles (replace, reduce, refine) regarding animal testings. On the contrary, we further concentrated on strengthening the focus of our work by adding even more mechanistic insights and sincerely hope that this is acceptable!

Reviewer #2 (Remarks to the Author):

In this manuscript, Tschuck et al., report the results of the pharmacological high throughput screen aiming to identify novel ferroptosis inhibitors. The study reports a protective effect of Turofexorate and Fexaramide, two Farnesoid X Receptor (FXR) agonists. The authors propose

that the mechanism is mediated by the an FXR-dependent upregulation of several anti-ferroptotic proteins including GPX4, FSP1, SCD1 and PPARα. The study is exciting and the effect of FXR as a central regulator of ferroptosis is important and clinically relevant. The observation that the agonist triggers a strong response in FSP1 expression level in primary hepatocytes is particularly exciting. I only have a few remarks the authors might wish to address to increase the soundness of their work-

We thank this referee for the very positive and encouraging feedback.

1. I would strongly encourage the authors to check if FSP1 is also upregulated at the protein level in their model. If available, testing the other regulators would be also advisable.

We treated HepG2 cells with the FXR agonists Turofexorate and Fexaramine and observed similar results on protein levels compared to the mRNA levels for selected target genes: upon treatment, the key anti-ferroptotic regulators FSP1 and GPX4 were upregulated on the protein level (**NEW Supplementary Fig. 3B**). The same effect could be seen in FXR-overexpressing HT-1080 cells, where increasing protein levels of FXR led to elevated protein levels of GPX4 and FSP1 (**NEW Supplementary Fig. 3A**). Western Blot bands were quantified using ImageJ by normalizing to β -Actin.

2. To clearly demonstrate the on-target effect of Turo/Fexa the generation of FXR knockout cell line would be important to unequivocally link the agonist to its target in the context of ferroptosis. Its not unlikely that the response could be mediated independent of FXR, which is fine – and could stimulate subsequent studies to full characterize this.

To strengthen our hypothesis that the observed anti-ferroptotic effect is indeed mediated by FXR and its downstream activity, we created a HepG2 FXR knockout cell line using CRISPR-Cas9. The knockout was verified by qRT-PCR (**NEW Supplementary Fig. 2A**). Knockout of FXR sensitized HepG2 cells to ferroptosis induction (**NEW Figure 3D**). Moreover, Turofexorate and Fexaramine failed to rescue RSL3-induced ferroptosis in FXR-KO cells (**NEW Figure 3E**). The KO data are in agreement with FXR inhibition (Figure 3F).

Interestingly, during the generation of FXR KO cells, we noticed that the FXR-KO HepG2 cells grew very poorly, and we suspect that cells with a complete knockout probably died during the process of expansion. This may be a further indication that FXR is important in averting oxidative and ferroptotic stress in hepatocytes. Together, the observed effects in the generated HepG2 FXR-KO cells are very convincing and suggests an on-target effect of both Turofexorate and Fexaramine.

As an additional proof, we overexpressed FXR in HT-1080 cells and treated them with FXR inhibitor Guggulsterone (FXRi). We observed that the upregulation of all investigated anti-ferroptotic target genes (FSP1, GPX4, PPAR α , ACSL3, SCD1) was reverted to control levels after treatment with FXRi (**NEW Supplementary Figure S4**). This shows that the upregulation of ferroptosis-inhibitory genes is directly linked to FXR activity as a transcription factor to activate target genes.

3. Similarly, it would be advisable to exclude unwanted antioxidant activity of Turo/Fexa (a simple DPPH assay should suffice).

To exclude any radical scavenging or iron chelating activity of Turofexorate and Fexaramine, we performed several cell-free assays: a cell-free BODIPY-C11 assay and a DPPH assay showed that both compounds are no antioxidants, and an iron chelator assay with EDTA as a positive control showed no iron chelating capacity (**NEW Supplementary Fig. 2B-D**), further indicating that Turofexorate and Fexaramine have on-target activity.

4. Are canonical FXR-inducible genes associated with ferroptosis resistance in other cell lines? Some exploration of databases such as the CTRP or Depmap could offer additional insights.

According to Fiorucci et al. (Fiorucci *et al.*, 2007), canonical FXR target genes are: NR0B2 (SHP), CYP7A1, Abcb11, FGF15, OST α/β , Fabp6, Bsep, Mrp2, UGT2B4, PLTP, ApoA-I, ApoC-III and others. Our database research in CTRP and Depmap did not yield definitive results about connections of these genes to ferroptosis. However, since FXR and bile acid metabolism are also connected to lipid metabolism, some of the canonical target genes may have an influence on lipid composition affecting ferroptosis. We added this information to the results and discussion sections. Yet, based on our data, we are confident that the most pronounced effect of FXR to inhibit ferroptosis is through transcriptional upregulation of FSP1, GPX4, PPAR α , ACSL3, and SCD1.

Reviewer #3 (Remarks to the Author):

These elegant series of studies indicate that FXR/RXR activation results in decrease in lipid peroxidation and inhibition of ferroptosis.

We thank this referee for the very positive feedback.

This reviewer has some questions:

1. Please present Figure 4F (Lipidomics) in a clearer format and explain the lipids that are regulated and their significance.

In the revised version, we changed **Figure 4F** and rearranged the lipids according to their headgroups (PC, PE, etc.) for more clarity. The relative abundance of the measured lipid class is displayed as a heatmap with the coloring: blue for low abundance, and red for high abundance. Lipid classes with similar abundances throughout the treatments were clustered together.

We included the full names for the headgroups into the figure legend. The number behind the colon indicates the number of double bonds present in the acyl tail.

When analyzing the lipidomics data, we focused not on single lipids, but instead on different groups of lipids and their changes between the different treatments. Especially relevant for ferroptosis induction are poly-unsaturated fatty acids (PUFAs). We could observe that upon RSL3 treatment the analyzed PUFA-PLs decreased due to degradation. Upon treatment with the FXR agonist Turofexorate, the lipid composition was reverted closer to control level (lipid block

on the right) or even improved (lipid block on the left). The differences between the positive control-treated samples (Ferrostatin-1) and Turofexorate-treated samples can be explained by the different mechanism of action of the two compounds: Fer-1 is a strong antioxidant and directly inhibits lipid peroxidation, whereas Turofexorate activates FXR, which in turn upregulates expression of antioxidant regulators or lipid-modulating regulators, e.g., PPAR α .

2. Do these lipid changes also occur *in vivo*, in models of liver injury associated with ferroptosis?

We agree that it would be interesting to see if our observed changes in lipid composition during ferroptosis induction and rescue by FXR activation could be extrapolated to *in vivo* models. Unfortunately, the lipid composition varies from cell to cell and between species. In particular, the length of an acyl chain and the number of double bonds can be different. Thus, it is not possible to make *in vivo* prediction for these exact lipids.

3. Is there any evidence that inhibition of ferroptosis with ferrostatin-1 or liproxstatin-1 or FXR agonists prevent lipid peroxidation and ferroptosis *in vivo*?

The ferroptosis inhibitor Ferrostatin-1 (Fer-1) is unstable in plasma and therefore cannot be used *in vivo*. However, Fer-1 could be optimized into the molecule 16-86 that was effective to reduce ferroptosis in a murine kidney Ischemia-Reperfusion Injury (IRI) model (Linkermann *et al.*, 2014). Also, the optimized Fer-1-derived analog UAMC-3203 showed *in vivo* efficacy in a mouse model of acute iron poisoning (Hofmans *et al.*, 2016; Devisscher *et al.*, 2018).

Liproxstatin-1, another ferroptosis inhibitor, was tested in a mouse model of renal and hepatic IRI and showed good *in vivo* potency to inhibit ferroptosis (Friedmann Angeli *et al.*, 2014).

Kim *et al.* recently demonstrated that FXR activation is able to attenuate cisplatin-induced acute kidney injury (AKI) in a mouse model (Kim *et al.*, 2022).

These studies and many more demonstrate that *in vivo* inhibition of ferroptosis can be of therapeutic value. Some of these points are now added to the introduction or discussion.

4. Which of the 5 genes stated in Figure 5B are most specific for ferroptosis?

GPX4 is the central enzyme that counteracts ferroptotic cell death (Hadian and Stockwell, 2020; Stockwell, 2022). GPX4 uses GSH to revert lipid hydroperoxides into their alcohol forms. Inducible knockouts of GPX4 in murine kidneys led to premature death due to massive ferroptosis (Friedmann Angeli *et al.*, 2014).

FSP1 was discovered in 2019 as an additional important ferroptosis suppressor (Bersuker *et al.*, 2019; Doll *et al.*, 2019). FSP1 inhibits ferroptosis independent of GPX4 activity.

ACSL3 is necessary to activate free fatty acids to fatty acyl-CoA, before they can be incorporated into the cell membrane as MUFA-PLs and exert their ferroptosis-inhibitory effects (Magtanong *et al.*, 2019). SCD1 is needed to synthesize MUFAs, since it catalyzes the formation of a double bond in the saturated fatty acid stearic acid (Tesfay *et al.*, 2019). MUFAs have been shown to have very potent anti-ferroptotic activity, which makes ACSL3 and SCD1 important regulators of ferroptosis (Hadian and Stockwell, 2020).

As a major transcriptional regulator of lipid synthesis and metabolism, PPAR α is involved in MUFA-PL synthesis. It also upregulates FSP1. Thus, it acts as an indirect ferroptosis inhibitor (Venkatesh *et al.*, 2020).

In conclusion, the most important genes among our target genes are GPX4 and FSP1, because they act as the major anti-ferroptotic regulators and, when knocked out, lead to ferroptotic cell death. The other three genes are also important regulators, but have a less central role in controlling ferroptosis.

5. In Figure 5F it would be desirable to compare effects of CDC and OC with FXR Overexpression as in Figure 5B.

In HepG2 cells we are not able to achieve an overexpression, as these cells already have high levels of FXR (**Fig. 3b, Supplementary Fig. 1**), and upregulation leads to a negative feedback loop that reduces the levels again. Thus, we are not able to compare target gene upregulation upon FXR activation versus OE. This is why we chose the HT-1080 with low FXR levels for OE.

References

- Bersuker, K., J.M. Hendricks, Z. Li, L. Magtanong, B. Ford, P.H. Tang, M.A. Roberts, B. Tong, T.J. Maimone, R. Zoncu, M.C. Bassik, D.K. Nomura, S.J. Dixon and J.A. Olzmann, 2019. The coq oxidoreductase fsp1 acts parallel to gpx4 to inhibit ferroptosis. *Nature*, 575(7784): 688-692. Available from <https://www.ncbi.nlm.nih.gov/pubmed/31634900>. DOI 10.1038/s41586-019-1705-2.
- Cao, L., R. Qin and J. Liu, 2023. Farnesoid x receptor protects against lipopolysaccharide-induced endometritis by inhibiting ferroptosis and inflammatory response. *Int Immunopharmacol*, 118: 110080. DOI 10.1016/j.intimp.2023.110080.
- Chong, H.K., A.M. Infante, Y.K. Seo, T.I. Jeon, Y. Zhang, P.A. Edwards, X. Xie and T.F. Osborne, 2010. Genome-wide interrogation of hepatic fxr reveals an asymmetric ir-1 motif and synergy with Irf-1. *Nucleic Acids Res*, 38(18): 6007-6017. DOI 10.1093/nar/gkq397.
- Devisscher, L., S. Van Coillie, S. Hofmans, D. Van Rompaey, K. Goossens, E. Meul, L. Maes, H. De Winter, P. Van Der Veken, P. Vandenabeele, T.V. Berghe and K. Augustyns, 2018. Discovery of novel, drug-like ferroptosis inhibitors with in vivo efficacy. *Journal of medicinal chemistry*, 61(22): 10126-10140. DOI 10.1021/acs.jmedchem.8b01299.
- Doll, S., F.P. Freitas, R. Shah, M. Aldrovandi, M.C. da Silva, I. Ingold, A. Goya Grocin, T.N. Xavier da Silva, E. Panzilius, C.H. Scheel, A. Mourao, K. Buday, M. Sato, J. Wanninger, T. Vignane, V. Mohana, M. Rehberg, A. Flatley, A. Schepers, A. Kurz, D. White, M. Sauer, M. Sattler, E.W. Tate, W. Schmitz, A. Schulze, V. O'Donnell, B. Proneth, G.M. Popowicz, D.A. Pratt, J.P.F. Angeli and M. Conrad, 2019. Fsp1 is a glutathione-independent ferroptosis suppressor. *Nature*, 575(7784): 693-698. Available from <https://www.ncbi.nlm.nih.gov/pubmed/31634899>. DOI 10.1038/s41586-019-1707-0.
- Fiorucci, S., G. Rizzo, A. Donini, E. Distrutti and L. Santucci, 2007. Targeting farnesoid x receptor for liver and metabolic disorders. *Trends Mol Med*, 13(7): 298-309. DOI 10.1016/j.molmed.2007.06.001.
- Friedmann Angeli, J.P., M. Schneider, B. Proneth, Y.Y. Tyurina, V.A. Tyurin, V.J. Hammond, N. Herbach, M. Aichler, A. Walch, E. Eggenhofer, D. Basavarajappa, O. Rådmark, S. Kobayashi, T. Seibt, H. Beck, F. Neff, I. Esposito, R. Wanke, H. Förster, O. Yefremova, M. Heinrichmeyer, G.W. Bornkamm, E.K. Geissler, S.B. Thomas, B.R. Stockwell, V.B. O'Donnell, V.E. Kagan, J.A. Schick and M. Conrad, 2014. Inactivation of the ferroptosis regulator gpx4 triggers acute renal failure in mice. *Nat Cell Biol*, 16(12): 1180-1191. DOI 10.1038/ncb3064.
- Hadian, K. and B.R. Stockwell, 2020. Snapshot: Ferroptosis. *Cell*, 181(5): 1188-1188 e1181. Available from <http://www.ncbi.nlm.nih.gov/pubmed/32470402>. DOI 10.1016/j.cell.2020.04.039.
- Hofmans, S., T. Vanden Berghe, L. Devisscher, B. Hassannia, S. Lyssens, J. Joossens, P. Van Der Veken, P. Vandenabeele and K. Augustyns, 2016. Novel ferroptosis inhibitors with improved potency and adme properties. *Journal of medicinal chemistry*, 59(5): 2041-2053. DOI 10.1021/acs.jmedchem.5b01641.
- Kim, D.H., H.I. Choi, J.S. Park, C.S. Kim, E.H. Bae, S.K. Ma and S.W. Kim, 2022. Farnesoid x receptor protects against cisplatin-induced acute kidney injury by regulating the transcription of ferroptosis-related genes. *Redox Biol*, 54: 102382. DOI 10.1016/j.redox.2022.102382.
- Linkermann, A., R. Skouta, N. Himmerkus, S.R. Mulay, C. Dewitz, F. De Zen, A. Prokai, G. Zuchtriegel, F. Krombach, P.S. Welz, R. Weinlich, T. Vanden Berghe, P. Vandenabeele, M. Pasparakis, M. Bleich, J.M. Weinberg, C.A. Reichel, J.H. Bräsen, U. Kunzendorf, H.J. Anders, B.R. Stockwell, D.R. Green and S. Krautwald, 2014. Synchronized renal tubular cell death involves ferroptosis. *Proc Natl Acad Sci U S A*, 111(47): 16836-16841. DOI 10.1073/pnas.1415518111.
- Magtanong, L., P.J. Ko, M. To, J.Y. Cao, G.C. Forcina, A. Tarangelo, C.C. Ward, K. Cho, G.J. Patti, D.K. Nomura, J.A. Olzmann and S.J. Dixon, 2019. Exogenous monounsaturated fatty acids promote a ferroptosis-resistant cell state. *Cell Chem Biol*, 26(3): 420-432 e429. Available from <https://www.ncbi.nlm.nih.gov/pubmed/30686757>. DOI 10.1016/j.chembiol.2018.11.016.

- Stockwell, B.R., 2022. Ferroptosis turns 10: Emerging mechanisms, physiological functions, and therapeutic applications. *Cell*, 185(14): 2401-2421. DOI 10.1016/j.cell.2022.06.003.
- Tesfay, L., B.T. Paul, A. Konstorum, Z. Deng, A.O. Cox, J. Lee, C.M. Furdui, P. Hegde, F.M. Torti and S.V. Torti, 2019. Stearoyl-coa desaturase 1 protects ovarian cancer cells from ferroptotic cell death. *Cancer Res*, 79(20): 5355-5366. DOI 10.1158/0008-5472.Can-19-0369.
- Venkatesh, D., N.A. O'Brien, F. Zandkarimi, D.R. Tong, M.E. Stokes, D.E. Dunn, E.S. Kengmana, A.T. Aron, A.M. Klein, J.M. Csuka, S.H. Moon, M. Conrad, C.J. Chang, D.C. Lo, A. D'Alessandro, C. Prives and B.R. Stockwell, 2020. Mdm2 and mdmx promote ferroptosis by ppar α -mediated lipid remodeling. *Genes Dev*, 34(7-8): 526-543. DOI 10.1101/gad.334219.119.
- Xing, G., L. Meng, S. Cao, S. Liu, J. Wu, Q. Li, W. Huang and L. Zhang, 2022. Ppar α alleviates iron overload-induced ferroptosis in mouse liver. *EMBO Rep*, 23(8): e52280. DOI 10.15252/embr.202052280.

REVIEWERS' COMMENTS

Reviewer #1 (Remarks to the Author):

The authors almost address my concerns.

Reviewer #2 (Remarks to the Author):

The authors have satisfactorily addressed my points -

I would only suggest that in Figure 3D and 3E the authors remove FXR KO found below "HepG2", given that the nomenclature WT and KO is already depicted on the "x" axis this becomes confusing.

Congratulations on this interesting and important work

Reviewer #3 (Remarks to the Author):

This reviewer thanks the authors for addressing my initial concerns and suggestions.

Reviewer #1 (Remarks to the Author):

The authors almost address my concerns.

Thanks for the acceptance

Reviewer #2 (Remarks to the Author):

The authors have satisfactorily addressed my points -

I would only suggest that in Figure 3D and 3E the authors remove FXR KO found below "HepG2", given that the nomenclature WT and KO is already depicted on the "x" axis this becomes confusing.

Congratulations on this interesting and important work

Thanks. We changed the labeling

Reviewer #3 (Remarks to the Author):

This reviewer thanks the authors for addressing my initial concerns and suggestions.

Thanks.